# REASONING TO EDIT: HYPOTHETICAL INSTRUCTION-BASED IMAGE EDITING WITH VISUAL REASONING

## ABSTRACT

Instruction-based image editing (IIE) has advanced rapidly with the success of diffusion models. However, existing efforts primarily focus on simple and explicit instructions to execute editing operations such as adding, deleting, moving, or swapping objects. They struggle to handle more complex implicit hypothetical instructions that require deeper reasoning to infer plausible visual changes and user intent. Additionally, current datasets provide limited support for training and evaluating reasoning-aware editing capabilities. Architecturally, these methods also lack mechanisms for fine-grained detail extraction that support such reasoning. To address these limitations, we propose **Reason50K**, a large-scale dataset specifically curated for training and evaluating hypothetical instruction–reasoning image editing, along with **ReasonBrain**, a novel framework designed to reason over and execute implicit hypothetical instructions across diverse scenarios. Reason50K includes over 50K samples spanning four key reasoning scenarios: Physical, Temporal, Causal, and Story reasoning. ReasonBrain leverages Multimodal Large Language Models (MLLMs) for editing guidance generation and a diffusion model for image synthesis, incorporating a Fine-grained Reasoning Cue Extraction (FRCE) module to capture detailed visual and textual semantics essential for supporting instruction reasoning. To mitigate the semantic loss, we further introduce a Cross-Modal Enhancer (CME) that enables rich interactions between the fine-grained cues and MLLM-derived features. Extensive experiments demonstrate that ReasonBrain consistently outperforms state-of-the-art baselines on reasoning scenarios while exhibiting strong zero-shot generalization to conventional IIE tasks.

## 1 INTRODUCTION

The successful deployment of diffusion models (Sohl-Dickstein et al., 2015; Ho et al., 2020) in text-to-image (T2I) generation (Ramesh et al., 2022; Rombach et al., 2022b; Saharia et al., 2022; He et al., 2024) has significantly accelerated the development of *instruction-based image editing* (IIE) (Nguyen et al., 2024a). IIE focuses on performing precise and localized modifications on a source image in response to human commands, thereby enhancing the controllability and accessibility of visual manipulation (Fu et al., 2024). Prior works typically utilize the CLIP-based text encoder inherent in T2I diffusion models for instruction embedding (Brooks et al., 2023; Zhang et al., 2023; Hui et al., 2024; Zhao et al., 2024; Geng et al., 2024; Zhang et al., 2024), which is insufficient for comprehending complex instructions. To address this limitation, recent methods (Tian et al., 2025; Huang et al., 2024; Nguyen et al., 2024b; Wang et al., 2024b; Li et al., 2024; Zhou et al., 2025; Sun et al., 2025) have proposed substituting it with multimodal large language models (MLLMs) (Touvron et al., 2023; Liu et al., 2024b), enabling richer cross-modal understanding and better alignment with user intent. However, despite the significant progress achieved by these efforts, the following limitations remain underexplored:

(L1) **Overlooking hypothetical instructions.** Existing IIE methods are primarily designed to handle simple, explicit, and goal-directed instructions (e.g., "remove the dog"), which typically correspond to straightforward editing operations such as adding, replacing, or deleting objects. However, users often begin without a clear editing objective and instead express ambiguous intent through hypothetical instructions (e.g., "What would happen if the ice cube was left in the sun?"). As illustrated in Fig. 1, current methods struggle to interpret and act on such inputs. Successfully ad-

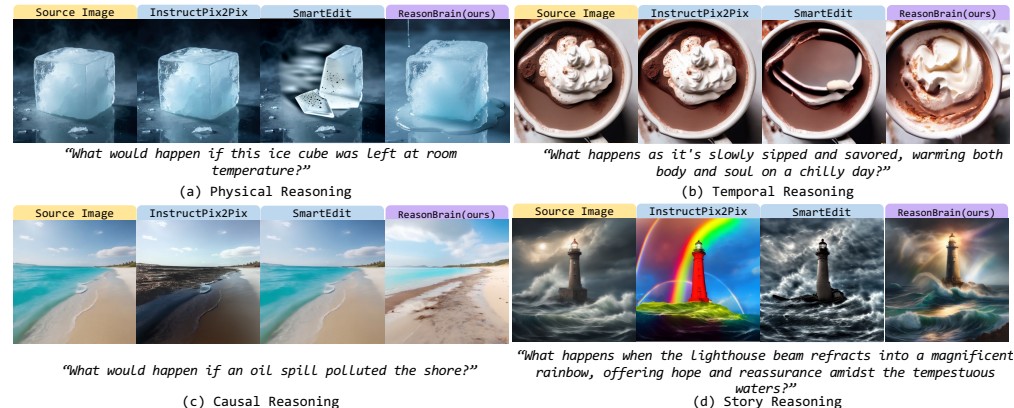

Figure 1: Current efforts fail to handle hypothetical instructions, producing incorrect results, while our method generates plausible, reasoning-aware edits.

dressing these cases requires models to go beyond surface-level edits and perform deeper reasoning about real-world context, physical changes, and the causal or temporal implications of the instruction. Moreover, currently there is no dataset specifically designed for hypothetical instruction-based editing that offers sufficient scale and scenario diversity (Huang et al., 2024; Yang et al., 2024; Jin et al., 2024; Meng et al., 2024).

(L2) **Insufficient reasoning ability.** While integrating MLLMs (Touvron et al., 2023; Liu et al., 2024b) can enhance instruction comprehension and improve alignment with user commands to some extent, these models still lack dedicated mechanisms for deep reasoning over hypothetical instructions (see Fig. 1 results of SmartEdit). We attribute this limitation to the prevailing paradigm's reliance on coarse-grained features extracted directly from the input image and instruction, which fail to capture the fine-grained semantic cues necessary for implicit reasoning or fully exploit the MLLM's embedded world knowledge (Wang et al., 2024a). For instance (Fig. 1(a)), reasoning over such an instruction requires jointly interpreting both visual and textual cues. Visually, elements like the cube's sharp edges, surface gloss, and surrounding environment reveal its physical state and context. Textually, features such as conditional phrasing, object references, and temporal expressions together suggest a melting process.

To address these limitations, we propose a unified solution: a large-scale hypothetical instruction-based dataset, **Reason50K**, and a tailored reasoning-aware framework, **ReasonBrain**. Reason50K comprises diverse hypothetical instructions spanning four reasoning categories: physical, temporal, causal, and story-based reasoning, totaling 51,039 samples. Each sample consists of a source image, a hypothetical instruction, and a corresponding target image that reflects the intended edit. ReasonBrain is a hybrid framework that jointly and interactively performs reasoning and editing, thereby overcoming the need for multi-round refinement of textual instructions (Fu et al., 2024). This unified design mitigates reasoning uncertainty and reduces inference time. Specifically, ReasonBrain consists of an MLLM and a diffusion model, augmented with two specialized modules for visual guidance, reasoning, and semantic enrichment: the Fine-grained Reasoning Cue Extraction (FRCE) module and the Cross-Modal Enhancer (CME). The FRCE module extracts detailed reasoning cues through two branches: the visual reasoning branch captures both local and global visual semantics to model spatial relationships and object-level interactions, while the textual reasoning branch identifies key object references and contextual intent from hypothetical instructions, enriched with relevant visual context. These fine-grained features, combined with multi-scale image tokens and textual embeddings, are input to the MLLM alongside learnable tokens to implicitly generate reasoning-aware visual guidance. Finally, the CME enhances these signals via semantic complementarity across modalities, producing well-aligned, semantically rich guidance for diffusion-based image editing. In sum, our contributions are as follows:

- We systematically extend traditional Instruction-Based Image Editing (IIE) to Hypothetical Instruction-Reasoning Image Editing (HI-IE). This task involves implicit, ambiguous, and hypothetical editing instructions that demand deeper reasoning over contextual cues, physical dynamics, and user intent.

- We curate a new large-scale dataset, **Reason50K**, specifically designed to support the reasoning of hypothetical instruction in image editing. It contains 51,039 triplets of source image, hypothetical instruction, and target image, covering four distinct reasoning categories: physical, temporal, causal, and story-based reasoning.
- We propose **ReasonBrain**, a novel image editing framework that combines a MLLM with fine-grained reasoning cues extraction and a cross-modal enhancer. Together, these components endow the model with implicit cross-modal reasoning capabilities, enabling it to infer plausible, knowledge-grounded transformations and produce semantically coherent guidance for diffusion-based image editing under complex hypothetical scenarios.
- We conduct extensive experiments on both Reason50K and widely used benchmark datasets, demonstrating the effectiveness and generalization ability of ReasonBrain across reasoning-intensive and standard understanding-based editing scenarios.

## 2 RELATED WORK

**Instruction-based Image Editing (IIE).** IIE (Brooks et al., 2023) aims to train generative models to manipulate a given image based on user-provided instructions. A milestone in this field is Instruct-Pix2Pix (IP2P) (Brooks et al., 2023), which is the first to incorporate natural language instructions into image editing by fine-tuning a text-to-image (T2I) diffusion model on paired image-instruction datasets. Subsequent works have built upon IP2P by introducing novel curated datasets to enhance real-world editing performance and generate high-quality outputs (Zhang et al., 2023; Hui et al., 2024; Zhao et al., 2024; Liu et al., 2024a; Yu et al., 2024). Others have focused on improving instruction-output alignment by incorporating advanced techniques such as reward learning (Zhang et al., 2024; Bai et al., 2024) and multi-task training (Sheynin et al., 2024). Recently, researchers have integrated multimodal large language models (MLLMs) (Touvron et al., 2023; Liu et al., 2024b) into existing image editing paradigms to enhance the model's ability to understand complex instructions. We refer to these approaches as MLLM-enhanced methods (Fu et al., 2024; Huang et al., 2024; Li et al., 2024; Wang et al., 2024b; Zhou et al., 2025; Tian et al., 2025). For instance, MGIE (Fu et al., 2024) leverages MLLMs to generate expressive instructions and provide explicit guidance, thereby enhancing editing performance. SmartEdit (Huang et al., 2024) further introduces a bidirectional interaction module that facilitates mutual understanding between the MLLM output and the input image. Despite recent advances, most existing efforts still rely on direct and explicit instructions (e.g., "Remove the ice"), limiting their ability to handle hypothetical instructions (e.g., "What would happen if the ice cube melted?") that require deeper reasoning. While MLLMs offer general world knowledge, current frameworks lack mechanisms to extract and utilize fine-grained reasoning details. Our **ReasonBrain** builds upon the MLLM-enhanced paradigm by introducing dedicated reasoning-aware modules for generating precise visual guidance, with an emphasis on accurately inferring the implicit intent and real-world context embedded in hypothetical instructions.

**Reasoning-aware Datasets for Image Editing.** Only a few works have explored reasoning-aware datasets for image editing (Huang et al., 2024; Yang et al., 2024; Jin et al., 2024). ReasonEdit (Huang et al., 2024) is designed primarily for evaluating the reasoning capabilities of image editing models and contains only a small set of textual samples, making it insufficient for training. Moreover, it focuses on object-level reasoning based on explicit instructions, while overlooking reasoning about editing operations themselves. EditWorld (Yang et al., 2024) aims to inject physical dynamics simulation capabilities into models across both realistic and virtual scenarios, but does not emphasize instruction-level reasoning. ReasonPix2Pix (Jin et al., 2024) introduces indirect instructions in a descriptive, goal-oriented style but lacks the depth and diversity needed for more advanced hypothetical reasoning. In contrast to these prior efforts, we curate **Reason50K**, a large-scale dataset specifically tailored for hypothetical instruction reasoning, enabling models to understand and execute complex edits grounded in physical, temporal, causal, and narrative scenarios.

## 3 METHODOLOGY

The goal of our work is to perform image editing by reasoning from a user-provided *hypothetical instruction*–an implicit, often ambiguous prompt that requires the model to infer the intended transformation through deeper reasoning about real-world context, physical dynamics, or potential

outcomes (e.g., "What would happen if the sun were setting?"). We refer to this task as **Hypothetical Instruction-Reasoning Image Editing (HI-IE)**. To this end, we propose a unified solution comprising a large-scale dataset, **Reason50K**, specifically curated to support the challenging task of hypothetical HI-IE, and a reasoning-aware image editing framework, **ReasonBrain**.

### 3.1 REASON50K FOR REASONING INJECTING

To support our proposed HI-IE task, we construct a novel large-scale dataset Reason50K specifically curated to inject *hypothetical instruction-reasoning ability* into image editing models. Reason50K contains 51,039 samples, each consisting of an input image, a corresponding hypothetical instruction, and a target edited image. Unlike existing datasets that primarily feature explicit or goal-oriented prompts (e.g., "Remove the animal in the mirror" or "Make it snowy"), our instructions are implicit, open-ended, and reasoning-driven (e.g., "What would happen if the ice cube were left at room temperature?" or "What would this bouquet look like if it were split into two separate rose bouquets?"). This shift introduces a significantly higher level of abstraction and demands real-world understanding. Reason50K is constructed using an inverse-style strategy, where the source image is generated from the target. Specifically, we leverage GPT (Achiam et al., 2023) to produce hypothetical instructions based on user-provided prompts, and employ a diffusion model to generate multiple candidate source images. A hybrid scoring and evaluation scheme is then applied to select the most appropriate image, forming the final image pairs. Moreover, Reason50K is organized around four distinct reasoning scenarios: **Physical Reasoning**, **Temporal Reasoning**, **Causal Reasoning**, and **Story Reasoning**, each illustrated with representative examples and instructions in Fig. 2.

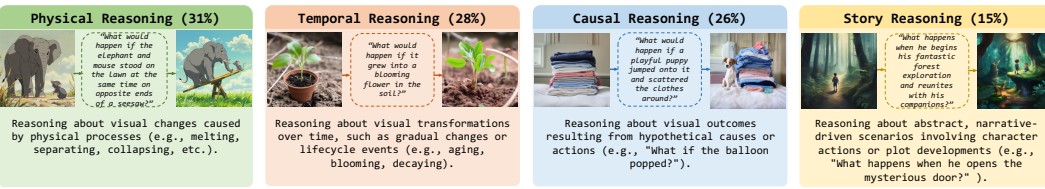

Figure 2: Reasoning scenarios in Reason50K. The percentages in parentheses indicate the proportion of each category. The text below each sample shows an instruction of the corresponding reasoning type.

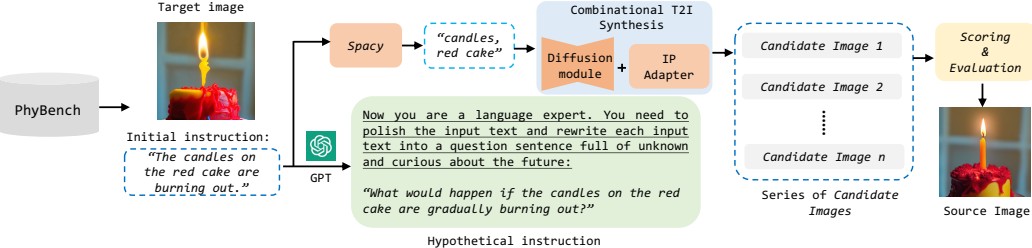

Figure 3: Data generation process.

**Dataset Generation** The construction of our data consists of two parts. The first part (over 90% of the entire dataset) is generated following the pipeline illustrated in Fig. 3, which adopts an inverse generation strategy–deriving the source image from the target. Specifically, we first adopt the same procedure as PhyBench (Meng et al., 2024) to generate target images along with their initial instructions. Each initial instruction is then rewritten into a hypothetical form using prompt-based rewriting with GPT (Achiam et al., 2023). In parallel, we use SpaCy[1] to perform named entity recognition (NER) on the initial instruction to extract candidate objects for source image generation. These candidates are passed to a diffusion model equipped with an IP-Adapter to synthesize multiple image variants. Each candidate image is subsequently evaluated by GPT, and the top-N images are selected based on a combination of GPT scores and perceptual quality metrics (He et al., 2023; Hore & Ziou, 2010; Zhang et al., 2018). Each selected image, along with the corresponding

---

[1] https://spacy.io/

hypothetical instruction and target image, constitutes a sample in our dataset. The second part (less than 10%) consists of story-type samples derived from EditWorld (Yang et al., 2024). We apply the same instruction rewriting, scoring, and filtering process to refine and select high-quality samples for inclusion.

Table 1: Comparison of different datasets. Note that since ReasonPix2Pix (Jin et al., 2024) is not publicly available, we reference sample cases from their paper and exclude it from further experimental comparison (see Fig. 6).

| Datasets | Reasoning Instruction | Automatic Generated | Open Domain | #Edits | #Editing Types | Source Example | Instruction | Target Example |
|---|---|---|---|---|---|---|---|---|
| ReasonPix2Pix (Jin et al., 2024) | ✓ | ✓ | ✗ | 40,212 | – | | *A colorful insect has landed* | |
| EditWorld (Yang et al., 2024) | ✗ | ✓ | ✓ | 8,674 | 7 | | *Shifted her gaze to the left.* | |
| ReasonEdit (Huang et al., 2024) | ✗ | ✓ | ✗ | 219 | – | | *Please remove the empty plate.* | |
| **Reason50K (Ours)** | ✓ | ✓ | ✓ | **51,039** | **4** | | *What would happen if the cat stood in front of a mirror?* | |

**Reason50K vs. Existing Datasets.** Tab. 1 summarizes the key differences between our dataset and existing ones. In comparison: 1) *ReasonPix2Pix* (Jin et al., 2024) enhances instruction reasoning by using LLMs to rewrite explicit editing commands into goal-oriented implicit instructions. However, the dataset primarily consists of descriptive instructions and lacks more complex, abstract hypothetical instructions that require deeper contextual and causal reasoning. In addition, it lacks systematic categorization. 2) *EditWorld* (Yang et al., 2024) focuses on simulating world dynamics across both real and virtual scenarios. Although it includes some hypothetical instructions, its primary objective is to inject physical and temporal simulation capabilities into image editing models, rather than equipping them with implicit reasoning skills for understanding hypothetical instructions. Additionally, the overall scale of the EditWorld dataset is significantly smaller than ours. 3) *ReasonEdit* (Huang et al., 2024) is a small-scale dataset designed primarily for evaluation. It focuses on object-level reasoning from explicit instructions and lacks both diversity and editing operation reasoning. Most importantly, it is not sufficient for model training. In contrast, our **Reason50K** is the first to provide *systematic, large-scale support* for training and evaluating *hypothetical instruction reasoning* across *diverse scenarios*. Each instruction requires deep reasoning grounded in contextual understanding and world knowledge—spanning physical, causal, temporal, and story-based scenarios—to guide image editing. This goes beyond surface-level transformations or explicit object manipulation. By incorporating carefully crafted hypothetical instructions, the dataset significantly promotes deeper semantic reasoning within image editing models.

It is noted that our dataset Reason50K is constructed from synthetic data, as it is difficult to obtain image pairs from videos that support hypothetical editing with clear reasoning structures. In addition, there is currently no standardized benchmark or evaluation protocol for the frame-by-frame assessment of reasoning-based edits. Nevertheless, the generation process of Reason50K is carefully designed to ensure both semantic consistency and high visual quality, thereby enabling the synthetic dataset to effectively support generalizable research.

## 3.2 REASONBRAIN

ReasonBrain comprises an MLLM for visual guidance, reasoning, and a diffusion model responsible for conditional image generation. To support knowledge reasoning, we incorporate a Fine-Grained Reasoning Cues Extraction (FRCE) module. In addition, we introduce a Cross-Modal Enhancer (CME) to further enrich semantic representations through modality-specific refinement. The overall framework of ReasonBrain is illustrated in Fig. 4.

**Fine-grained Reasoning Cues Extraction.** Existing efforts utilize $\mathcal{E}_I(I)$ and $\mathcal{E}_T(H)$ directly for visual guidance generation, overlooking fine-grained cues critical for implicit reasoning, such as local object attributes, subtle spatial relationships, and context-dependent semantics between image regions and instructions, etc. To address this limitation, we introduce a Fine-Grained Reasoning

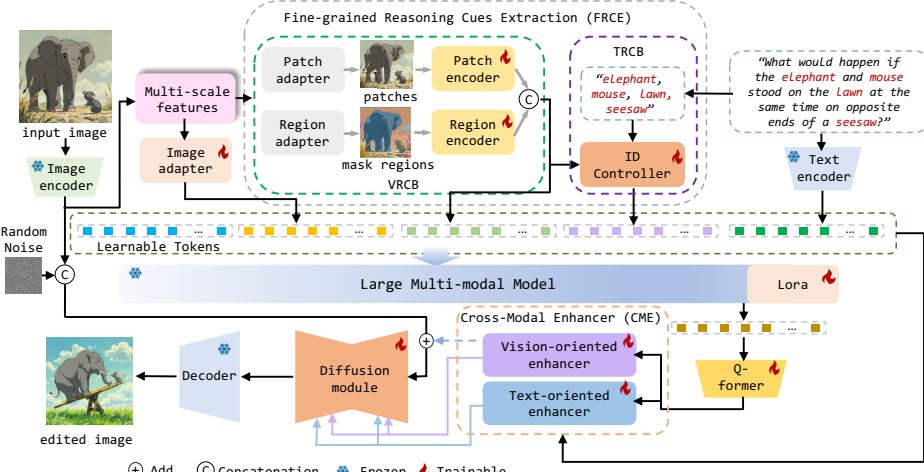

Figure 4: The overall framework of ReasonBrain. Given an input image $I$ and a hypothetical instruction $H$, ReasonBrain first encodes them into multi-scale visual features and textual tokens using the image encoder $\mathcal{E}_I(\cdot)$ and text encoder $\mathcal{E}_T(\cdot)$, respectively. These features are then passed into the FRCE module to learn detailed reasoning cues. Subsequently, all learned features are fed into the MLLM to generate visual guidance, which is further transformed via a QFormer to align with the diffusion model's latent space. Finally, the resulting visual guidance interacts with the previously extracted fine-grained cues through a CME module to enhance semantic representation, which is then used to condition the diffusion model for final image generation.

Cues Extraction (FRCE) module, which comprises two specialized branches designed to capture fine-grained visual and textual reasoning cues, respectively.

*(1) Visual Reasoning Cues Branch (VRCB).* This branch aims to extract fine-grained visual cues from both local and global perspectives. Specifically, the *local perspective* focuses on capturing object parts, textures, and spatial patterns, which are essential for implicit reasoning tasks involving fine object distinctions and subtle appearance changes. Inspired by the MAE framework (He et al., 2022), we first divide the visual features $\mathcal{E}_I(I)$ into patches using a patch adapter $\mathcal{P}(\cdot)$, and then apply a patch-level feature extractor $E_{\mathcal{P}}(\cdot)$ to obtain localized visual representations. This process is formalized as: $R_{local} = E_{\mathcal{P}}(\mathcal{P}(\mathcal{E}_I(I)))$. In contrast, the *global perspective* captures inter-object relationships and contextual information across the entire scene, which are beneficial for reasoning tasks that require an understanding of object interactions, event causality, and broader scene dynamics. We first employ an instance segmentation model (e.g., SAM (Kirillov et al., 2023; Ravi et al., 2024)) to segment objects from the background in the image. Subsequently, a region-level feature extractor $E_{\mathcal{R}}(\cdot)$ is trained to learn holistic semantic representations for each segmented instance. The entire process is formalized as:

$$R_{global} = E_{\mathcal{R}}(\text{SAM}(\mathcal{E}_I(I))). \tag{1}$$

After this dual-level operation, we concatenate $R_{local}$ and $R_{global}$ to form the final visual reasoning features $R_V$ used for subsequent processing.

*(2) Textual Reasoning Cues Branch (TRCB).* This branch aims to extract the key object referenced in $H$, serving as a bridge between linguistic intent and visual reasoning. Specifically, we first employ GPT (Achiam et al., 2023) to extract the referenced object from the instruction and serialize it into a structured object token $O$. We then introduce an *ID Controller* to enhance the model's ability to perform object-grounded reasoning by facilitating interaction between the object token and the visual reasoning features $R_V$. This module not only enriches the object token with visual context, enabling the model to reason about the object beyond its textual description, but also aligns the linguistic reference with its corresponding visual entity, helping to resolve ambiguities and prevent semantic drift during generation. The architecture of the ID Controller is illustrated in Fig. 5(a) and is implemented using a cross-attention layer (Lin et al., 2022) followed by a feed-forward network, formally defined as:

$$R_T = \text{FF}(\text{Cross-Atten}(R_V, O)). \tag{2}$$

Subsequently, following prior works (Huang et al., 2024; Zhou et al., 2025), we concatenate $r$ additional learnable tokens $\mathcal{Q} = \{[\text{IMG}_1], \ldots, [\text{IMG}_r]\}$ with the extracted features and feed them into the MLLM for guidance generation. This operation transforms the implicit reasoning process into a token prediction task, where the MLLM learns to generate the embeddings of these $r$ tokens, which serve as visual editing guidance. The process is formalized as:

$$\mathcal{T} = [\text{IA}(\mathcal{E}_I(I)), R_V, R_T, \mathcal{E}_T(H), \mathcal{Q}], \quad V = \text{MLLM}(\mathcal{T}; \theta). \tag{3}$$

Here, $[\cdot, \cdot]$ denotes concatenation, and $\text{IA}(\cdot)$ is a trainable image adapter that maps visual features into the MLLM's latent space. The output $V \in \mathbb{R}^{r \times d}$ represents the hidden embeddings of the $r$ learnable tokens. In addition, we employ a QFormer (Li et al., 2023) to align the feature space between the MLLM and the diffusion model, defined as $\hat{V} = \text{QFormer}(V)$.

**Cross-Modal Enhancer.** To compensate for the potential loss of visual and textual details in $\hat{V}$, we introduce a Cross-Modal Enhancer (CME). The CME consists of a *visual-oriented enhancer* and a *textual-oriented enhancer*, both implemented using the same bidirectional interaction mechanism. Specifically, each enhancer comprises five hybrid cross-attention blocks, followed by a linear projection layer and a normalization layer (see Fig. 5(b)). These modules reuse intermediate features generated both before and after the MLLM layer, enabling fine-grained semantic enhancement within each

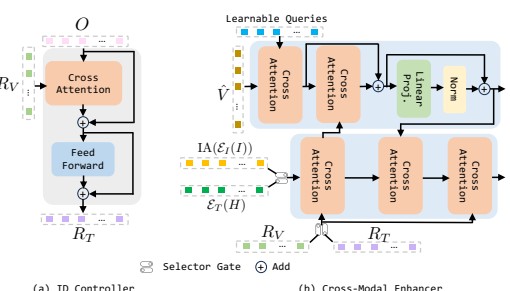

Figure 5: Network design of the (a) ID Controller and (b) Cross-Modal Enhancer.

modality. Here, we illustrate the process using the visual-oriented enhancer. First, a set of learnable tokens $Q$[2] interacts with $\hat{V}$ (as key and value) via a cross-attention block to produce $F_1$. In parallel, $\mathcal{E}_I(I)$ (query) attends to $R_V$ in a second cross-attention block, yielding $F_2$. Next, $F_1$ (query) and $F_2$ (key and value) are fused via a third block, followed by a residual connection and normalization to obtain $\bar{V}$. This refined representation then serves as key and value in a fourth block, interacting with $F_2$ (query). The output is further refined through a final cross-attention with $R_V$, producing the enhanced visual representation $\bar{R}_V$. Similarly, the textual-oriented enhancer is implemented by replacing $\mathcal{E}_I(I)$ and $R_V$ with $\mathcal{E}_T(H)$ and $R_T$, respectively. Finally, the CME module outputs four enhanced features, which are passed to the diffusion model together with $\mathcal{E}_I(I)$ and a noisy latent $z$ for final image generation. Due to space limitations, the training objectives of ReasonBrain are provided in App. C.

## 4 EXPERIMENTS

### 4.1 QUANTITATIVE RESULTS

**Performance Comparison on Reasoning Scenarios.** The experimental results on Reason50K are summarized in Tab. 2, covering four distinct types of reasoning scenarios. ReasonBrain consistently outperforms all SOTA baselines across all metrics, demonstrating superior ability to infer hypothetical instructions and produce logically accurate edits aligned with world knowledge. While UltraEdit and PixWizard achieve competitive performance, their advantage mainly stems from exposure to massive datasets with diverse instructions rather than true reasoning capabilities. Notably, MLLM-enhanced methods such as MGIE and SmartEdit still underperform, indicating that simply integrating MLLMs or fine-tuning on reasoning datasets is insufficient without mechanisms for extracting fine-grained reasoning cues. These results further suggest that scaling data alone cannot bridge the reasoning gap without dedicated architectural modules and training objectives. Additionally, we select three representative methods for evaluation on ReasonEdit and EditWorld. As shown in Fig. 6, ReasonBrain, trained solely on our Reason50K, generalizes effectively to novel reasoning scenarios and achieves the best overall performance across datasets.

**Generalization Comparison on Understanding Scenarios.** As shown in Tab. 3, Reason-Brain achieves the best overall performance among all SOTA methods, demonstrating strong zero-shot generalization to new datasets. This result also highlights that, although trained solely on

---

[2]The newly introduced symbols are used only in this section to illustrate the visual-oriented enhancer.

Table 2: Results on the Reason50K dataset for ReasonBrain and selected baselines. ↓ indicates that lower values are better, while ↑ indicates that higher values are better. The best results are highlighted in **bold**, and the second-best results are underlined.

| Method | Physical Reasoning | | | Temporal Reasoning | | | Causal Reasoning | | | Story Reasoning | | | Total | | |
|---|---|---|---|---|---|---|---|---|---|---|---|---|---|---|---|
| | CLIP↑ | MLLM↑ | Ins-Align↑ | CLIP↑ | MLLM↑ | Ins-Align↑ | CLIP↑ | MLLM↑ | Ins-Align↑ | CLIP↑ | MLLM↑ | Ins-Align↑ | CLIP↑ | MLLM↑ | Ins-Align↑ |
| InstructPix2Pix (Brooks et al., 2023) | 0.083 | 0.825 | 0.211 | 0.207 | 0.846 | 0.678 | 0.153 | 0.785 | 0.191 | 0.196 | 0.628 | 0.225 | 0.160 | 0.771 | 0.326 |
| MagicBrush (Zhang et al., 2023) | 0.102 | 0.844 | 0.335 | 0.228 | 0.877 | 0.725 | 0.162 | 0.802 | 0.344 | 0.209 | 0.635 | 0.359 | 0.175 | 0.790 | 0.441 |
| MGIE (Fu et al., 2024) | 0.098 | 0.802 | 0.288 | 0.213 | 0.832 | 0.685 | 0.155 | 0.761 | 0.328 | 0.201 | 0.622 | 0.322 | 0.167 | 0.754 | 0.406 |
| SmartEdit (Huang et al., 2024) | 0.118 | 0.849 | 0.602 | 0.226 | 0.881 | 0.779 | 0.165 | 0.823 | 0.385 | 0.211 | 0.655 | 0.361 | 0.180 | 0.802 | 0.532 |
| UltraEdit (Zhao et al., 2024) | 0.156 | 0.861 | 0.584 | 0.231 | 0.922 | 0.826 | 0.193 | 0.869 | 0.482 | 0.209 | 0.669 | 0.458 | 0.197 | 0.830 | 0.588 |
| PixWizard (Lin et al., 2024) | 0.161 | 0.881 | 0.481 | 0.234 | 0.951 | 0.833 | 0.132 | 0.863 | 0.393 | 0.172 | 0.697 | 0.389 | 0.175 | 0.848 | 0.524 |
| ReasonBrain (ours) | 0.186 | 0.902 | 0.846 | 0.267 | 0.977 | 0.894 | 0.282 | 0.891 | 0.858 | 0.301 | 0.736 | 0.798 | 0.259 | 0.877 | 0.847 |

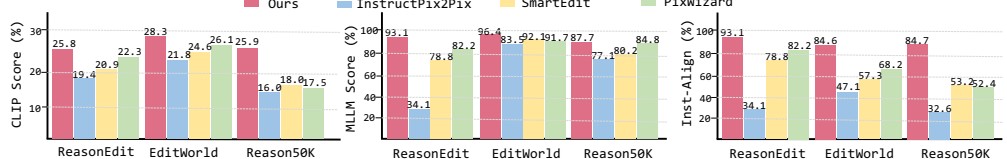

Figure 6: Results on ReasonEdit, EditWorld, and Reason50K for ReasonBrain and selected SOTA methods, highlighting performance under various reasoning datasets.

Table 3: Results on Emu Edit and MagicBrush test set for ReasonBrain and selected baselines.

| Method | Emu Edit Test set | | | | | MagicBrush Test Set | | | | |
|---|---|---|---|---|---|---|---|---|---|---|
| | $CLIP_{dir}$↑ | $CLIP_{im}$↑ | $CLIP_{out}$↑ | L1↓ | DINO↑ | $CLIP_{dir}$↑ | $CLIP_{im}$↑ | $CLIP_{out}$↑ | L1↓ | DINO↑ |
| InstructPix2Pix (Brooks et al., 2023) | 0.078 | 0.834 | 0.219 | 0.121 | 0.762 | 0.115 | 0.837 | 0.245 | 0.093 | 0.767 |
| MagicBrush (Zhang et al., 2023) | 0.090 | 0.838 | 0.222 | 0.100 | 0.776 | 0.123 | 0.883 | 0.261 | 0.058 | 0.871 |
| MGIE (Fu et al., 2024) | 0.083 | 0.746 | 0.231 | 0.163 | 0.594 | 0.116 | 0.745 | 0.251 | 0.162 | 0.577 |
| SmartEdit (Huang et al., 2024) | 0.092 | 0.858 | 0.274 | 0.119 | 0.771 | 0.119 | 0.895 | 0.262 | 0.094 | 0.820 |
| Emu Edit (Sheynin et al., 2024) | 0.109 | 0.859 | 0.231 | 0.094 | 0.819 | 0.135 | 0.897 | 0.261 | 0.052 | 0.879 |
| UltraEdit (Zhao et al., 2024) | 0.107 | 0.844 | 0.283 | 0.071 | 0.793 | - | 0.868 | - | 0.088 | 0.792 |
| PixWizard (Lin et al., 2024) | 0.104 | 0.845 | 0.248 | 0.069 | 0.798 | 0.124 | 0.884 | 0.265 | 0.063 | 0.876 |
| ReasonBrain (ours) | 0.126 | 0.923 | 0.302 | 0.051 | 0.898 | 0.139 | 0.928 | 0.281 | 0.049 | 0.893 |

hypothetical instructions, ReasonBrain retains a robust understanding of clear, goal-directed editing commands. Furthermore, the competitive performance of Emu Edit, UltraEdit, and PixWizard suggests that scaling up with large and diverse datasets can further enhance model effectiveness on conventional instruction-based image editing tasks.

## 4.2 QUALITATIVE RESULTS

As shown in Fig. 7, we visualize editing results of ReasonBrain and selected SOTA methods across four different reasoning scenarios. Our method demonstrates superior capability in executing hypothetical editing instructions by accurately reasoning about user intent, affected objects, and their plausible state transitions, while also maintaining stability in non-edited regions (original scene/object identity (ID)). For instance, in the first row (physical reasoning), our method successfully generates a physically plausible and contextually coherent scene–depicting the elephant and mouse standing on opposite ends of a seesaw, where the heavier elephant naturally tilts the seesaw downward. Additionally, our ReasonBrain preserves the original lawn, seesaw structure, and object proportions. This demonstrates that our model can not only reason about relative weight, spatial arrangement, and the physical dynamics implied by the instruction, but also maintain a certain degree of identity consistency. In contrast, InstructPix2Pix fails to capture the core intent, producing an irrelevant result. Other methods may include the correct objects mentioned in the instruction–such as the elephant, mouse, and seesaw–but fall short in modeling their physical interactions, resulting in unrealistic or semantically inconsistent compositions. Notably, ChatGPT-4o produces an implausible outcome in which the mouse outweighs the elephant, contradicting basic physical intuition. We attribute this failure to hallucinations introduced by the language model (Huang et al., 2025).

Moreover, although ReasonBrain may introduce slight scene adjustments compared to the source image, these changes are strictly bounded by the hypothetical instruction. They serve as logical supplements to the core modification rather than arbitrary alterations, and non-target regions (e.g., walls, lawns) remain visually identical to the original. For example, in the third row (casual reasoning), ReasonBrain adjusts the camera distance and local contrast to make the bee–flower interaction visible. This is a necessary scene–level refinement to truly convey the hypothetical event. If one were to rigidly forbid any background adaptation, the generated bees would be barely visible or visually unnatural. In contrast, baselines (e.g., InstructPix2Pix cannot model the environmental in-

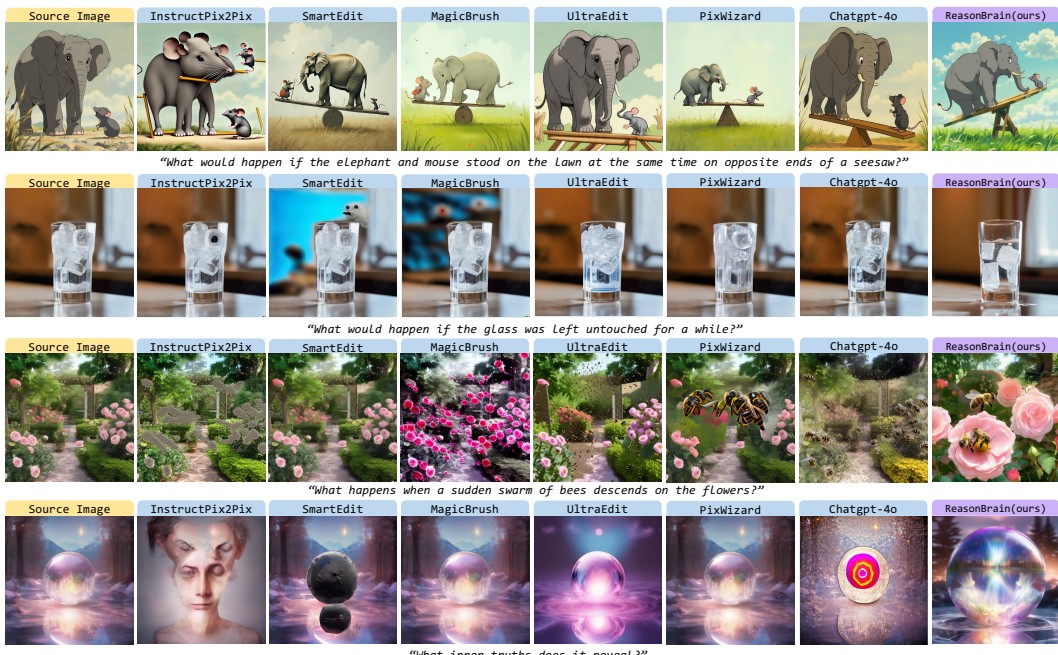

Figure 7: Qualitative comparison on Reason50K between ReasonBrain and selected SOTA methods. Compared to other SOTA methods, ReasonBrain demonstrates a strong ability to reason over implicit hypothetical instructions and produce semantically plausible edits grounded in world knowledge.

terplay between the bees and the flowers) preserve the overall scene appearance but fail to express the required interaction, illustrating that high appearance consistency does not imply correct reasoning. ReasonBrain strikes a balance between identity preservation and instruction expressiveness. Core ID elements (e.g., flower clusters, lawn layout, and non-target objects) remain intact, while subtle background refinements (such as camera distance, mild lighting, or contrast adjustments) are applied only when necessary to ensure the hypothetical change is visually coherent and semantically meaningful.

A similar pattern appears in the last row (story reasoning), where the instruction requires uncovering subtle, implicit transformations of the subject (e.g., hidden textures or symbolic details). Baselines such as InstructPix2Pix produce irrelevant outputs lacking reasoning, while SmartEdit fails to highlight implicit cues due to overly rigid background preservation. ReasonBrain, by contrast, achieves a balanced result in which the core identity (subject shape, global layout, and non-target regions) is retained, and only minimal, reasoning-driven background adjustments (e.g., localized light enhancement or slight contrast tuning) are applied to make the "inner truths" visually perceptible. These refinements are not excessive but necessary. Without them, the instruction-induced changes would remain imperceptible, undermining the purpose of implicit hypothetical editing.

## 4.3 ABLATION AND ANALYSIS

Table 4: Results of the ablation study on each component of ReasonBrain.

| | with Patch Branch | with Region Branch | with ID Controller | with Vision Enhancer | with Text Enhancer | CLIP Score↑ | MLLM Score↑ | Ins-Align Score↑ |
|---|---|---|---|---|---|---|---|---|
| ReasonBrain | × | × | × | × | × | 0.163 | 0.752 | 0.388 |
| | ✓ | × | × | × | × | 0.187 | 0.786 | 0.466 |
| | ✓ | ✓ | × | × | × | 0.206 | 0.802 | 0.529 |
| | ✓ | ✓ | ✓ | × | × | 0.239 | 0.833 | 0.758 |
| | ✓ | ✓ | ✓ | ✓ | × | 0.251 | 0.865 | 0.822 |
| | ✓ | ✓ | ✓ | × | ✓ | 0.248 | 0.845 | 0.785 |
| | ✓ | × | ✓ | ✓ | ✓ | 0.231 | 0.838 | 0.776 |
| | ✓ | × | ✓ | ✓ | ✓ | 0.246 | 0.847 | 0.788 |
| | × | ✓ | ✓ | ✓ | ✓ | 0.240 | 0.842 | 0.781 |
| | ✓ | ✓ | ✓ | ✓ | ✓ | **0.259** | **0.877** | **0.847** |

**Effectiveness of ReasonBrain's Components.** To evaluate the contribution of each component in ReasonBrain, we conduct ablation experiments by progressively integrating key modules. As shown in Tab. 4 and Fig. 8, the fine-grained visual features help the model capture subtle visual cues essential for reasoning. The ID Controller significantly improves performance by preserving object identity during cross-modal alignment, which is critical for accurate instruction reasoning. Additionally, the CME module enhances overall generation quality by reinforcing modality-specific semantics and providing more detailed guidance. Furthermore, we also observe that removing any individual component leads to a performance drop across all three metrics, indicating that each module contributes meaningfully to the overall effectiveness of the proposed framework.

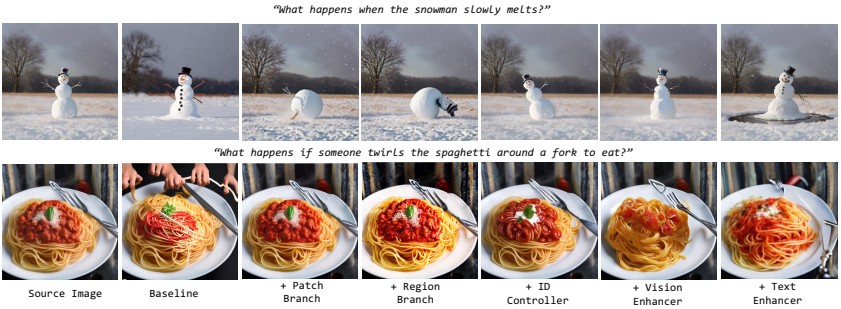

Figure 8: Qualitative comparison of ablation variants in ReasonBrain.

**Impact of Each Visual Branch in VRCB.** As shown in Fig. 9, we visualize the individual and combined contributions of the patch and region branches in our VRCB. The patch branch captures local details such as texture variations and appearance changes, which are useful for modeling subtle transformations (e.g., wind effects on kite fabric or shifting crop colors). In contrast, the region branch focuses on global semantic structures and object-level understanding, such as the overall layout of the kite or the boundary of the burning field. When used independently, each branch captures only partial aspects of the intended edit. The patch branch may introduce local distortions without

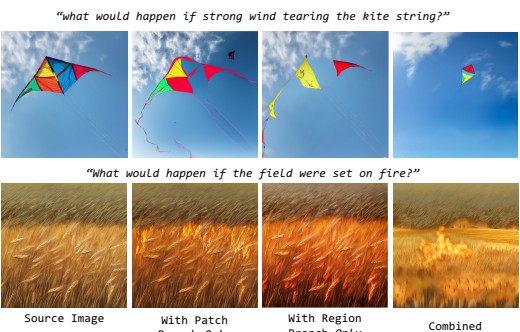

Figure 9: Qualitative comparison of patch and region branches in VRCB.

preserving object integrity, while the region branch may lack detailed variation. Their combination enables complementary integration of local precision and global semantics, resulting in more coherent, semantically accurate, and visually plausible edits under hypothetical instructions.

## 5 CONCLUSION

We extend instruction-based image editing to a hypothetical instruction reasoning setting and propose a unified solution from both dataset and model perspectives. Specifically, we curate Reason50K, a large-scale dataset of 51,039 samples specifically designed to support hypothetical instruction reasoning across four diverse categories: physical, temporal, causal, and story reasoning. Simultaneously, we introduce ReasonBrain, a novel framework that enhances instruction reasoning by combining a MLLM and a fine-grained feature extraction module. We further integrate a cross-modal enhancer to enrich the semantics of the guidance used for diffusion-based editing. We conduct extensive experiments on both reasoning-intensive and conventional understanding scenarios, demonstrating that ReasonBrain exhibits strong reasoning capabilities as well as robust generalization performance. In addition, our Reason50K can facilitates broader advancements in reasoning-aware image generation, providing an extensible resource for future research in this emerging direction.

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

APPENDIX

## A    REPRODUCIBILITY STATEMENT

We have already elaborated on all the models or algorithms proposed, experimental configurations, and benchmarks used in the experiments in the main body or appendix of this paper. Furthermore, we declare that the entire code used in this work will be released after acceptance.

## B    THE USE OF LARGE LANGUAGE MODELS

We use large language models solely for polishing our writing, and we have conducted a careful check, taking full responsibility for all content in this work.

## C    TRAINING OBJECTIVES OF REASONBRAIN

The training of ReasonBrain comprises two components: fine-tuning the MLLM and optimizing the diffusion model. Specifically, for fine-tuning the MLLM, we freeze most of its parameters and apply Low-Rank Adaptation (LoRA) (Hu et al., 2022) for efficient adaptation. The objective is defined as:

$$\mathcal{L}_{MLLM} = -\sum_{i=1}^{r} \log p_{\{\theta \cup \theta_{LoRA}\}} \left( [\text{IMG}_i] \mid \text{IA}(\mathcal{E}_I(I)), R_V, R_T, \mathcal{E}_T(H), [\text{IMG}_1], \dots, [\text{IMG}_{i-1}] \right), \tag{A1}$$

where $\theta_{LoRA}$ denotes the trainable parameters introduced by LoRA. This loss minimizes the negative log-likelihood of predicting each learnable token $[\text{IMG}_i]$ conditioned on the fine-grained features and previously predicted tokens. For image generation, we adopt a latent diffusion objective:

$$\mathcal{L}_{DM} = \mathbb{E}_{\mathcal{E}_I(\hat{I}), \mathcal{E}_I(I), H, \epsilon, t} \left[ \left\| \epsilon - \epsilon_\delta \left( t, [z_t, \mathcal{E}_I(I)] + \bar{R}_{visual} + \bar{R}_{text}, [\bar{e}_{visual}, \bar{e}_{text}] \right) \right\|_2^2 \right], \tag{A2}$$

where $\epsilon \sim \mathcal{N}(0,1)$ is the sampled noise and $z_t$ is the noisy latent at timestep $t$. $\epsilon_\delta(\cdot)$ denotes the denoising network trained to predict the added noise based on the timestep and the provided visual reasoning guidance. The overall training objective is defined as the sum of the MLLM and diffusion losses: $\mathcal{L} = \mathcal{L}_{MLLM} + \mathcal{L}_{DM}$. Moreover, to mitigate hallucination risks, we adopted a context- and instruction-aware token selection strategy inspired by SID (Huo et al., 2025), combined with the dynamic token propagation mechanism from TAME (Tang et al., 2025) in the training process of MLLM.

## D    SETUPS: DATASETS, METRICS, AND DETAILS

**Datasets**: We use Reason50K for both training and evaluation. Specifically, for each reasoning category, 400 samples are randomly selected for validation, while the remaining samples are used for training. In addition, we assess the reasoning capability of ReasonBrain on two external benchmarks: *ReasonEdit* (Huang et al., 2024) and *EditWorld* (Yang et al., 2024). To evaluate generalization on conventional understanding scenarios, we further test on the *MagicBrush Test Set* (Zhang et al., 2023) and the *Emu Edit Test Set* (Sheynin et al., 2024).

**Metrics**: To evaluate performance under reasoning scenarios, we adopt three metrics: *CLIP Score* (Radford et al., 2021), *MLLM Score* (Yang et al., 2024), and *Instruction Alignment (Ins-Align)* (Huang et al., 2024). Here, CLIP Score measures the semantic similarity between the edited image and the expected output text using CLIP's image-text embeddings. MLLM Score employs an MLLM to assess instruction-following performance. Following Yang et al. (2024), we provide the input description, editing instruction, and output description along with the edited image to Video-LLaVA. The prompt is defined as: "The input description: [object Object], the editing instruction: [object Object], and the output description: [object Object]. Please evaluate if the given edited image has been successfully edited. If yes, return 1; if not, return 0." The final score is the average of model judgments across all samples. In addition, Ins-Align Score evaluates how well the edited image aligns with the given instruction. Following Huang et al. (2024), ten human annotators independently rated the outputs on the Reason50K dataset, and we report the average alignment score. For understanding scenarios, we adopt *L1 distance*, *CLIP image similarity*, *DINO similarity*, *CLIP text-image similarity*, and *CLIP text-image direction similarity* as evaluation metrics.

**Implementation Details**: During training, we adopt the pre-trained LLaVAv1.1-7B (Liu et al., 2024b) and QFormer (Li et al., 2023) and employ DeepSpeed (Aminabadi et al., 2022) Zero-2 to perform LoRA (Hu et al., 2021) fine-tuning, with rank and alpha of 8 and 16, respectively. Following (Huang et al., 2024; Fu et al., 2024), we expand the original LLM vocabulary with 32 new tokens, and the QFormer is composed of 6 transformer layers and 77 learnable query tokens. For the base editing model, we implement it with Flux (Labs, 2024) using FLUX.1-dev, which consists of 12B parameters. Models for other qualitative results are implemented using SD series (CompVis, 2022; SimianLuo, 2024; Rombach et al., 2022a; AI, 2022) with their original codebase. Our model is trained with a batch size of 16, and we use AdamW (Loshchilov & Hutter, 2017) optimizer to train the model with the weight decay as 1e-2 and the learning rate as 1e-3. All the experiments are conducted on 16 H20 GPUs. For a fair comparison, all baseline models are fine-tuned on the same training set used by ReasonBrain.

## E   MORE QUANTITATIVE RESULTS

**Functional group studies**: To clarify the pivotal components and avoid fragmented analysis, we re-organize the ablations into coarse-grained functional groups, focusing on the core reasoning–editing interaction. As shown in Tab. A1 and visualized in Fig. A1, the Baseline Group (MLLM + diffusion only) produces only minimal or unrelated changes, achieving the lowest Ins-Align Score of **0.388**. This demonstrates that simply combining an MLLM with a diffusion model is insufficient for understanding and executing implicit hypothetical instructions. The FRCE Core Group (Method IDs 2–3) forms the foundation of reasoning–editing interaction. Without the ID Controller (Method 2), the model produces semantically inconsistent outcomes—such as generating an additional intact egg while also showing partial cracking—indicating a loss of object identity and clear semantic drift. With the ID Controller added (Method 3, full FRCE Core), the model preserves the correct object identity and generates a more coherent "dropped-egg" outcome, where the same egg is broken in a physically plausible way. This raises the Ins-Align Score to 0.758, a 43.3% improvement, confirming that the ID Controller is crucial for binding reasoning cues to the correct object and preventing unintended identity changes. Finally, the CME Enhancement Group (Method 4) achieves the strongest overall performance (CLIP: **0.259**, MLLM: **0.877**, Ins-Align: **0.847**). As seen in the rightmost result of Fig. A1, CME produces a clean, physically plausible "dropped and splattered" egg consistent with the instruction. CME refines the cross-modal alignment between FRCE-derived cues and diffusion-based editing features, reducing mismatches between the intended reasoning and the resulting edits. It acts as a targeted enhancer rather than a standalone reasoning module. In summary, our framework's effectiveness relies on two pivotal components: (1) the FRCE Core, with the ID Controller as the key mechanism for linking reasoning to identity-preserving edits; and (2) the CME module, which further enforces cross-modal consistency and elevates the overall reasoning quality.

Table A1: Quantitative results for the functional groups of ReasonBrain.

| Functional Group | Method ID | with Patch Branch | with Region Branch | with ID Controller | with Vision Enchancer | with Text Enchancer | CLIP Score↑ | MLLM Score↑ | Ins-Align Score↑ |
|---|---|---|---|---|---|---|---|---|---|
| Baseline Group | 1 | × | × | × | × | × | 0.163 | 0.752 | 0.388 |
| FRCE Core Group | 2 | ✓ | ✓ | × | × | × | 0.206 | 0.802 | 0.529 |
| FRCE Core Group + ID | 3 | ✓ | ✓ | ✓ | × | × | 0.239 | 0.833 | 0.758 |
| CME Enhancement | 4 | ✓ | ✓ | ✓ | ✓ | ✓ | **0.259** | **0.877** | **0.847** |

**Computational costs & lightweight alternative**: Tab. A2 shows the inference time of our Reason-Brain alongside all baselines. We find that our model demonstrates strong reasoning capabilities while maintaining an inference time comparable to that of existing methods. To further accelerate the model, we explored a lightweight variant, **ReasonBrain-3B**, by adopting a smaller pretrained MLLM. The model settings of ReasonBrain and **ReasonBrain-3B** are listed in Tab. A4. The performance comparison across all four reasoning categories between ReasonBrain-3B and our full model, ReasonBrain-7B, is illustrated in Tab. A3. We observe that, compared to ReasonBrain-7B, ReasonBrain-3B is significantly faster but exhibits only a slight performance drop. Nevertheless, it still outperforms all baselines reported in Tab. 2. This indicates that even in instruction-based reasoning scenarios, a lightweight MLLM can not only accelerate inference but also effectively identify the editing target and execute precise edits, supported by its strong reasoning ability and rich world knowledge.

*"What happens if they accidentally drop it??"*

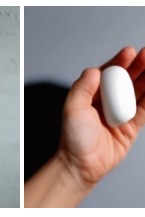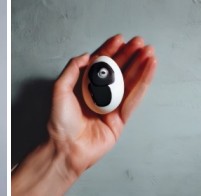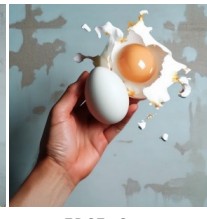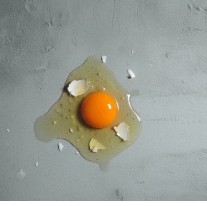

| Source Image | Baseline Group | FRCE Core Group | FRCE Core Group + ID | CME Enhancement |

Figure A1: Qualitative results for the functional component groups of ReasonBrain.

| Method | Inference Time (s) |
|---|---|
| InstructPix2Pix | 26 |
| MagicBrush | 28 |
| MGIE | 37 |
| SmartEdit | 33 |
| UltraEdit | 28 |
| PixWizard | 35 |
| **Ours** | **32** |

Table A2: Comparison of inference times across different methods.

| Model | CLIP ↑ | MLLM ↑ | Ins-Align ↑ | Inference Time (s) |
|---|---|---|---|---|
| ReasonBrain-3B | 0.238 | 0.852 | 0.822 | 24 |
| ReasonBrain-7B | 0.259 | 0.877 | 0.847 | 32 |

Table A3: Performance comparison of ReasonBrain and its lightweight variant.

**Multi-step editing**: We extend the original one-step experiments to include 2-step and 3-step editing scenarios (multi-hop reasoning chains, e.g., 'ice melting → water evaporating → damp ground forming'). The results (shown in Tab. A5) indicate that the model maintains robust performance across multiple editing steps, demonstrating the effectiveness of ReasonBrain in handling sequential multi-step reasoning-based editing. We attribute this to the ID Controller module, which operates independently at each editing step, enabling effective processing of chained instructions without causing significant feature drift.

**Human evaluation**: We randomly select 50 images corresponding to four distinct reasoning scenarios. For each image, we obtain the results of InstructPix2Pix, MagicBrush, MGIE, SmartEdit, UltraEdit and PixWizard. Then, randomly shuffle the order of these method results. For each set of images, we ask 30 participants to independently select the three best pictures. The first one is the best picture corresponding to the text prompt (i.e., Instruct Alignment), and the second one is the picture with the highest visual quality (i.e., Image Quality). The result is shown in Tab. A6. We found that 75.71% of participants believed ReasonBrain better reflected the correct reasoning behind the instructions, and 63.55% preferred the results generated by ReasonBrain. In contrast, all baseline methods received less than 10% on both evaluations.

**The influences of the data generation process on the final performance**: Let us briefly recall the key steps in our data generation process (see App. 3.1 and Fig. 3 for details):

(a) Utilizing PhyBench to acquire a target image and its initial prompt;

(b) Extracting candidate objects from the prompt (e.g., object identity, action, attributes);

(c) Using these candidate objects to guide a T2I model to generate candidate source images;

(d) Selecting the best candidate based on a combination of evaluation metrics;

(e) Rewriting the initial prompt into a hypothetical instruction format.

| Model Variant | MLLM Backbone | Diffusion Backbone | FRCE Params | CME Params | Total Trainable Params |
|---|---|---|---|---|---|
| ReasonBrain-7B | LLaVA-7B | SD 2.1 | 82M | 45M | 800M (LoRA-tuned) |
| ReasonBrain-3B | LLaVA-3B | SD 1.5 | 41M | 22M | 320M (LoRA-tuned) |

Table A4: Settings of ReasonBrain and its lightweight variant.

| Steps | CLIP ↑ | MLLM ↑ | Ins-Align ↑ |
|---|---|---|---|
| 1-step | 0.259 | 0.877 | 0.847 |
| 2-steps | 0.257 | 0.878 | 0.848 |
| 3-steps | 0.260 | 0.877 | 0.847 |

Table A5: Performance comparison under different editing steps.

| Method | Instruct-Alignment (%) | Image Quality (%) |
|---|---|---|
| InstructPix2Pix | 2.62 | 2.14 |
| MagicBrush | 3.11 | 5.65 |
| MGIE | 3.10 | 6.68 |
| SmartEdit | 3.55 | 6.85 |
| UltraEdit | 6.09 | 6.11 |
| PixWizard | 5.82 | 9.02 |
| **Ours** | **75.71** | **63.55** |

Table A6: Human evaluation results.

Finally, each data sample is composed of a selected candidate image, the rewritten instruction, and the original target image. We argue that the potential sources of influence on the final performance include:

- **Noise in Source Image Generation (Step a):** Some target images may contain distortions or semantically irrelevant content.

- **Missing Object Elements (Step b):** Key candidate objects may be omitted during prompt decomposition.

- **Lack of Constraints (Step c):** The T2I generation process does not explicitly enforce identity or background preservation when generating candidate source images.

- **Limited Evaluation Dimensions (Step d):** The scoring and selection criteria may overlook fine-grained visual details such as object consistency or scene coherence.

To this end, we conduct a preliminary ablation study to assess how these factors affect model performance. The results are summarized in Tab. A7. We find that when the initial step produces low-quality target images (error image), the model's performance drops significantly. Similarly, in the source image generation step, missing object elements (missing ID text) or ignoring ID correlations (w/o ID adapter) negatively impact performance. Finally, during the selection phase, using only a single metric (i.e., single selection) results in a considerable performance gap, highlighting the importance of applying a comprehensive set of evaluation metrics when constructing the final dataset.

**Identity preservation**: We followed the identity preservation metric proposed in (Yang et al., 2025) and conduct an evaluation across multiple baseline methods. The results are summarized in Tab. A8. As shown, our ReasonBrain achieves the highest score, indicating that it more effectively retains core elements from the source image while following the instruction. This demonstrates that our model trained on Reason50K is actually performing reasoning and not just learning a shortcut.

**Effectiveness of CME module**: To validate the effectiveness of the bidirectional information interaction in our proposed CME module, we conducted two comparative experiments. In *Exp 1*, we remove the CME module entirely and directly feed the feature output from QFormer into the diffusion model. This ablation study is designed to evaluate the effectiveness of the information interaction introduced by the CME module. In *Exp 2*, we aim to assess the necessity of bidirectional information interaction. Specifically, we retain only the cross-attention block on the image feature branch, discarding all other components of the CME module. As a result, the textual features from

| Setting | CLIP ↑ | MLLM ↑ | Ins-Align ↑ |
|---|---|---|---|
| Error Image | 0.104 | 0.425 | 0.226 |
| Missing ID Text | 0.125 | 0.465 | 0.233 |
| w/o ID Adapter | 0.087 | 0.312 | 0.158 |
| Single Selection | 0.172 | 0.705 | 0.505 |
| **Ours** | **0.259** | **0.877** | **0.847** |

Table A7: Impact of the data generation process on performance.

| Method | Identity Preservation ↑ |
|---|---|
| InstructPix2Pix | 5.13 |
| MagicBrush | 7.52 |
| MGIE | 7.63 |
| SmartEdit | 8.02 |
| UltraEdit | 5.68 |
| PixWizard | 8.55 |
| **Ours** | **9.72** |

Table A8: Comparison of identity preservation performance.

QFormer are applied to the image features in a unidirectional manner. The results are presented in Tab. A9, demonstrating the marginal gains brought by each component and highlighting the overall benefit of the complete CME design.

| Exp ID | Plain | Simple CA | CME | CLIP ↑ | MLLM ↑ | Ins-Align ↑ |
|---|---|---|---|---|---|---|
| 1 | ✓ | | | 0.239 | 0.833 | 0.758 |
| 2 | | ✓ | | 0.241 | 0.841 | 0.766 |
| Ours | | | ✓ | **0.259** | **0.877** | **0.847** |

Table A9: Ablation study on different design choices.

**Testing on different mapping settings**: We conduct an additional ablation study by extending our original o*ne-to-one mapping* to *one-to-two* and *one-to-three* mappings. The results are shown in Tab. A10, we find that incorporating multiple targets (i.e., one-to-two/one-to-three mappings) led to significant performance degradation across all metrics. These settings introduced training instability due to conflicting optimization signals from multiple targets for the same input. In particular, Ins-Align and Identity Preservation dropped significantly, indicating that using multiple targets for the same input makes it difficult to retain key elements from the source image–further emphasizing the importance of maintaining a one-to-one structure in our task.

**Framework design validation:** We conduct additional experiments to further validate the design of our ReasonBrain, particularly the bidirectional interaction between reasoning and editing within a unified framework. Specifically, we evaluate the FRCE module and the CME module independently. The FRCE module extracts fine-grained reasoning cues (e.g., physical object weight, temporal changes), injects them into the MLLM's visual-guidance generation, and binds them to the diffusion model via the QFormer. In contrast, the CME module enables mutual alignment between MLLM-generated reasoning tokens and diffusion-based editing features. The corresponding results are provided in Tab. A11 and Tab. A12. Our results show that removing the FRCE–diffusion interaction leads to a 12.3% drop in Ins-Align Score and an 8.7% decrease in visual plausibility (human evaluation). Using CME without bidirectional interaction further reduces the Ins-Align Score by 4.1%. These findings demonstrate the necessity of both modules and highlight the importance of their bidirectional integration.

## F  LIMITATIONS

**Limitations on Real-world Data Collection.** The construction of our dataset Reason50K follows the common practice in the image editing community (e.g., EmuEdit (Sheynin et al., 2024) and GPT-Image-Edit (Wang et al., 2025)), relying primarily on synthetic data. While synthetic datasets

| Setting | CLIP ↑ | MLLM ↑ | Ins-Align ↑ | Identity Preservation ↑ |
|---|---|---|---|---|
| Ours (one-to-one) | 0.259 | 0.877 | 0.847 | 9.72 |
| one-to-two | 0.152 | 0.680 | 0.315 | 4.25 |
| one-to-three | 0.088 | 0.258 | 0.204 | 1.66 |

Table A10: Comparison of one-to-many editing settings.

| Method | CLIP Score (↑) | MLLM Score (↑) | Ins-Align Score (↑) |
|---|---|---|---|
| ReasonBrain (Full, FRCE + Diffusion Interaction) | 0.259 | 0.877 | 0.847 |
| ReasonBrain (w/o FRCE-Diffusion Binding, MLLM Only) | 0.228 | 0.801 | 0.743 |
| Performance Drop | -12.0% | -8.7% | -12.3% |

Table A11: Performance comparison: with vs. without FRCE–Diffusion interaction

| CME Mode | CLIP Score (↑) | MLLM Score (↑) | Ins-Align Score (↑) |
|---|---|---|---|
| Bidirectional | 0.259 | 0.877 | 0.847 |
| Unidirectional (Reasoning→Editing only) | 0.248 | 0.842 | 0.806 |

Table A12: Performance comparison: bidirectional vs. unidirectional CME

provide semantic clarity and high visual quality, they may not fully capture the complexities, artifacts, and temporal dynamics inherent in real-world video data. Collecting high-quality datasets from real-world video sources remains particularly challenging due to the scarcity of suitable image pairs with explicit reasoning logic and the absence of standardized benchmarks or evaluation protocols for reasoning-based edits. Moreover, prior efforts such as EditWorld (Yang et al., 2024) show that frames extracted from videos often suffer from low resolution and poor aesthetics, making them suboptimal for training high-fidelity models. Thus, addressing this gap by curating reasoning-aligned, high-quality video benchmarks constitutes an important direction for our future work.

## G  QUALITATIVE COMPARISON: REASONBRAIN VS. ADDITIONAL SOTA MODELS

As shown in Fig. A2, we compare the generation results of ReasonBrain against several more general models, including Bagel (Deng et al., 2025), Bagel-Thinking, Flux Kontext (Batifol et al., 2025), and Qwen-Image-Edit (Wu et al., 2025). It is evident that ReasonBrain consistently produces the most faithful results, achieving superior semantic alignment, more accurate reasoning that reflects the instruction-implied changes, and overall higher visual coherence.

## H  MORE QUALITATIVE RESULTS

To further demonstrate the editing performance of ReasonBrain compared to SOTA methods, we present additional qualitative results in Fig. A3 and Fig. A4, Fig. A5 and Fig. A6. It is evident that ReasonBrain consistently outperforms other SOTA approaches, producing more coherent and visually plausible edits. These results highlight ReasonBrain's ability to reason effectively from hypothetical instructions and generate outputs that closely align with the transformations implied by the underlying reasoning.

## I  BROADER IMPACT AND ETHICS STATEMENT

We plan to make the dataset and associated code publicly available for research. Nonetheless, we acknowledge the potential for misuse, particularly by those aiming to generate misinformation using our methodology. We will release our code under an open-source license with explicit stipulations to mitigate this risk.

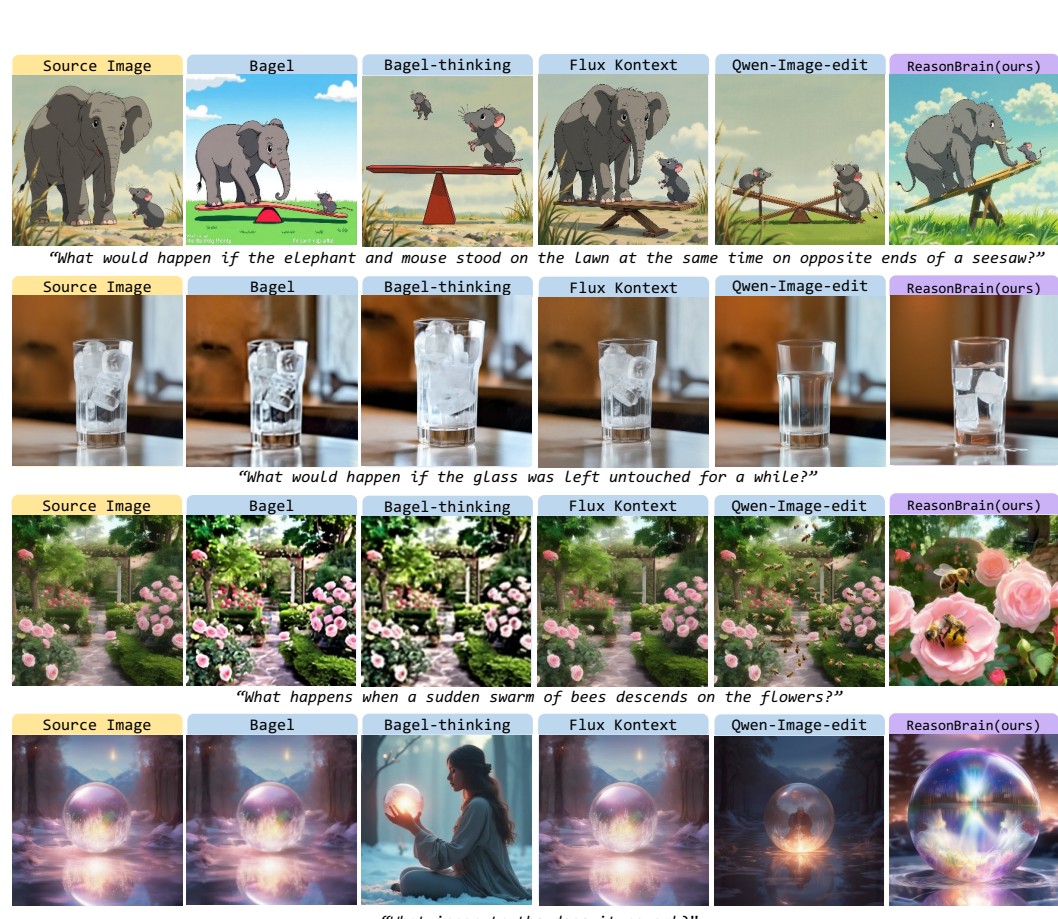

Figure A2: Qualitative comparison between ReasonBrain and additional SOTA models.

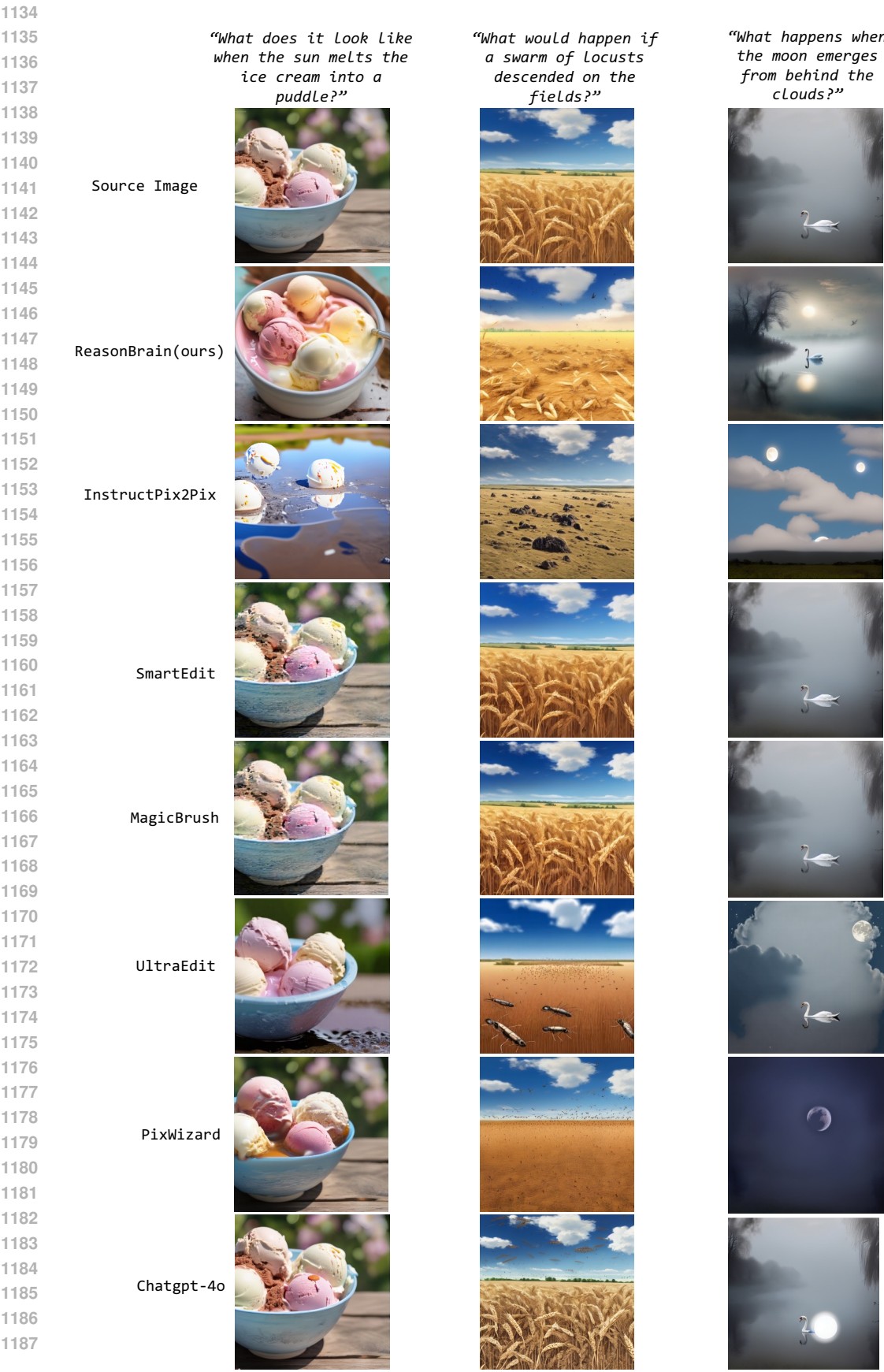

Figure A3: Qualitative comparison on Reason50K between ReasonBrain and selected SOTA methods.

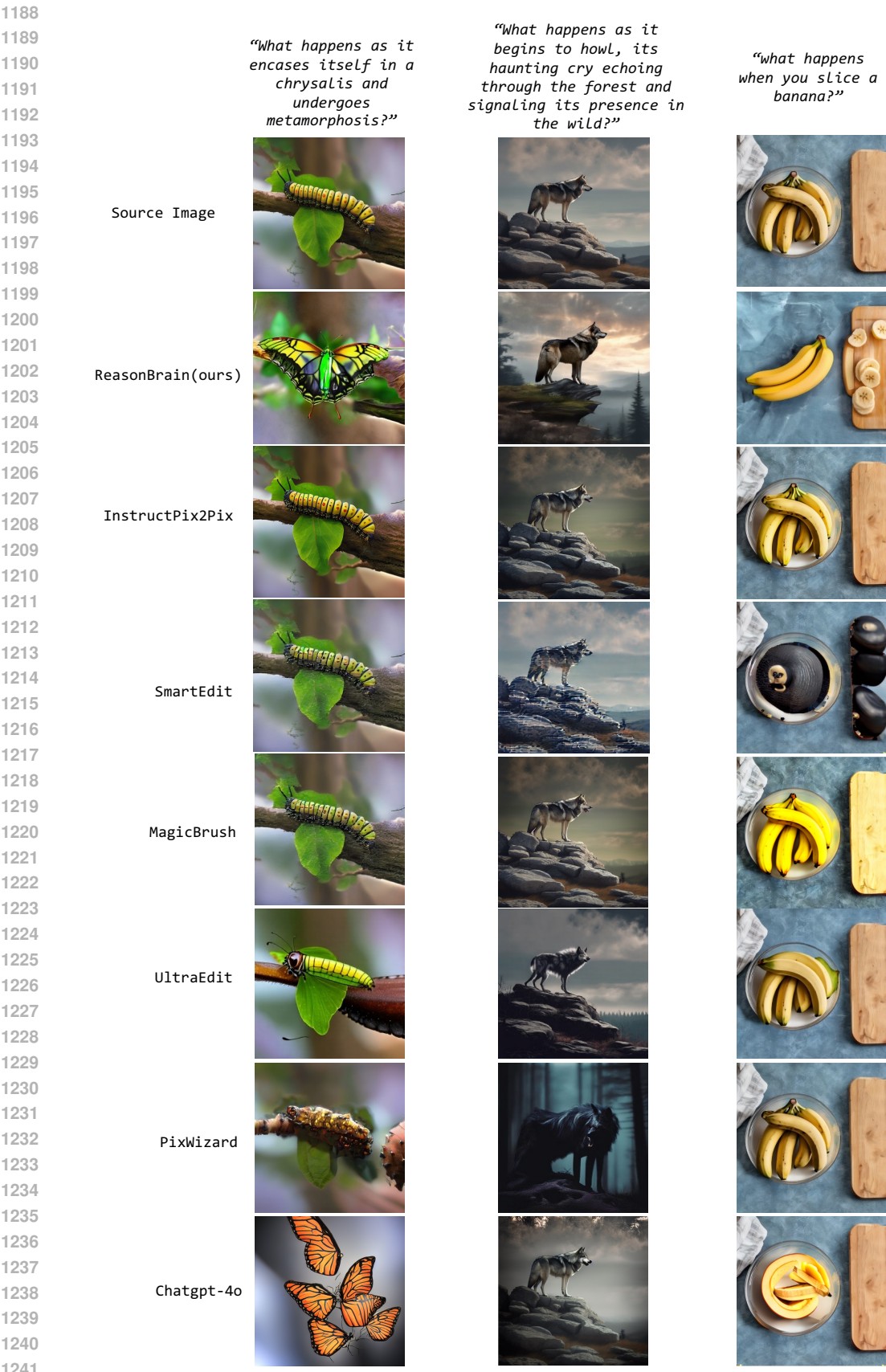

Figure A4: Qualitative comparison on Reason50K between ReasonBrain and selected SOTA methods.

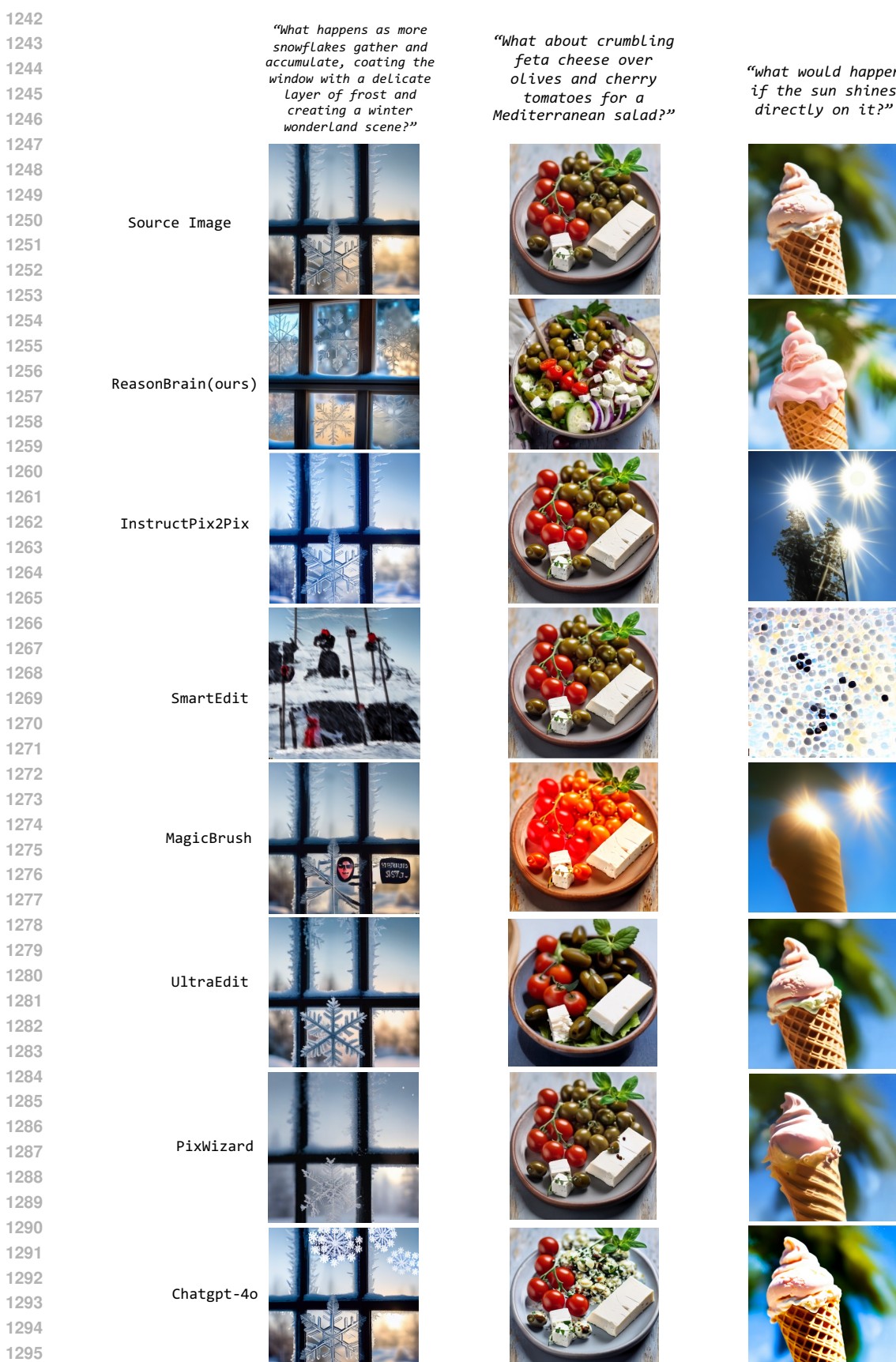

Figure A5: Qualitative comparison on Reason50K between ReasonBrain and selected SOTA methods.

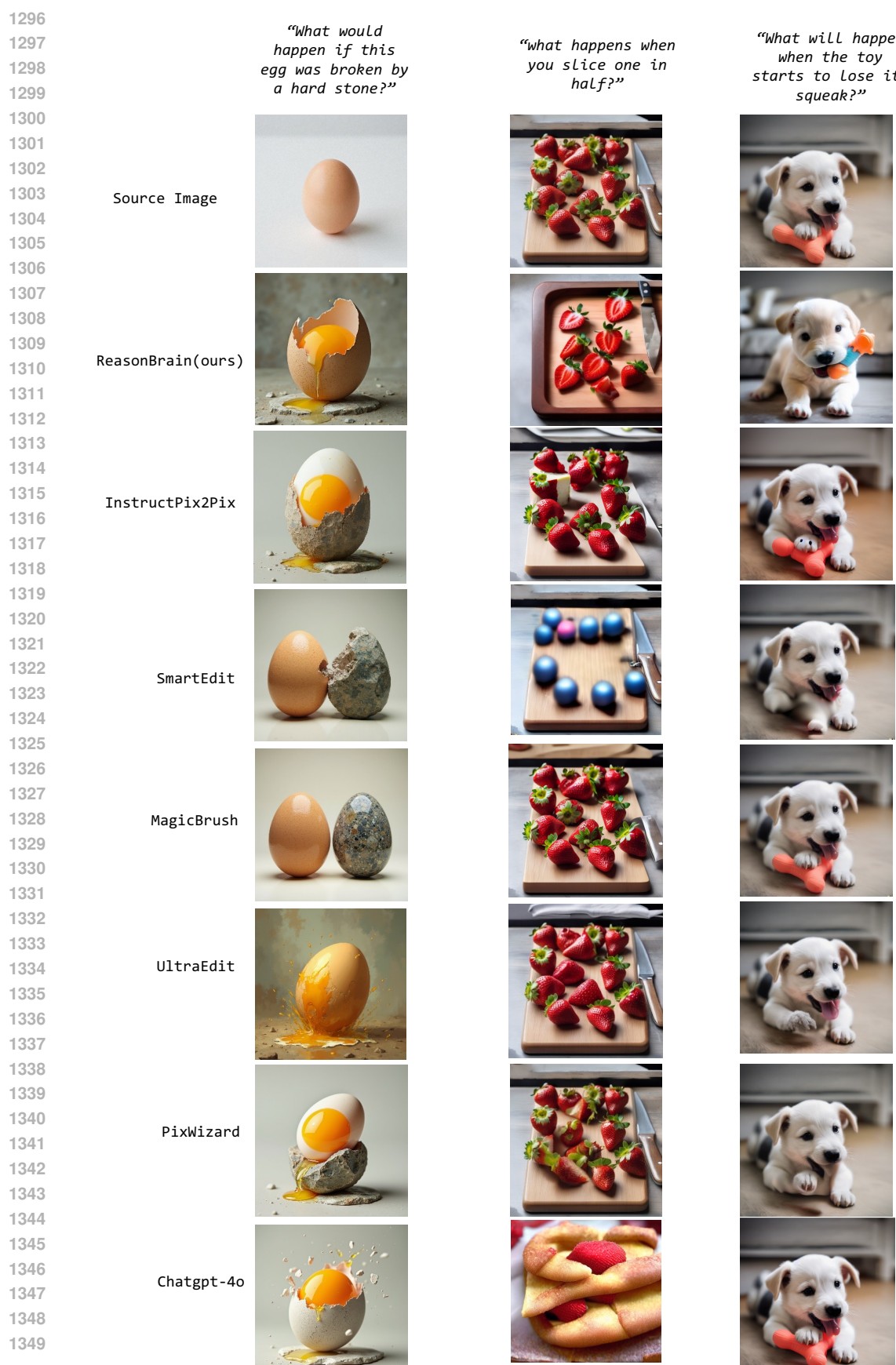

Figure A6: Qualitative comparison on Reason50K between ReasonBrain and selected SOTA methods.

