# OpenReview forum: "Reasoning to Edit: Hypothetical Instruction-Based Image Editing with Visual Reasoning"
_ICLR.cc/2026/Conference — Submitted to ICLR 2026_

### Official Review · Reviewer_FYWs · 2025-10-26

**Soundness:** 3
**Presentation:** 3
**Contribution:** 3
**Rating:** 6
**Confidence:** 3

**Summary:**

This paper focuses on hypothetical instruction–reasoning image editing, an interesting task. It proposes a large-scale dataset called Reason5k, with over 5k samples, covering 4 types of reasoning, physical, temporal, causal, and story reasoning. It also proposes a framework called ReasonBrain which includes a multi-modal large language model for instruction reasoning and a diffusion model for image editing. A Fine-grained Reasoning Cue Extraction (FRCE) module and a Cross-Modal Enhancer (CME) are introduced to facilitate reasoning. Extensive experiment results prove the effectiveness of the proposed method.

**Strengths:**

1. This paper is easy to read, with a clear organization. The figures demonstrate the proposed dataset and framework very vividly. Though it is not the first paper that targets on tackling reasoning instructions, I still think this is an interesting problem.
2. I think it is good to categorize different types of reasoning, as shown in this paper, they have physical, temporal, causal, and story reasoning (Figure 2). The dataset can make contributions to the community.
3. The results are extensive and impressive, proving the model's ability to tackle instruction-based image editing with reasoning. I like both the quantitative and qualitative results in the experiment section.

**Weaknesses:**

1. I like the dataset and I wonder that can we expand it to a more large-scaled one in an easy manner?
2. The novelty of the framework part of this paper is ok, but not very high to me. For example, the similar idea about incorporating a large language model to enhance reasoning ability has been proposed in [1]. I think the authors can make more discussions on the novelty of the model part during rebuttal, especially on how the reasoning part interact with the image editing part. There has been some discussions in the paper now, but I want to see more results, and I think this is what the novelty of the framework lies in.
[1] LISA: Reasoning Segmentation via Large Language Model

3. Can you can have some discussions or comparison with models such as Qwen-image?
[2] Qwen-image technical report, 2025a. URL https://arxiv.org/abs/2508.02324
Now I think this paper is ok, but if the authors can address the concerns here, it will be a better one.

**Questions:**

See weaknesses, especially the second point.

---

> ### Author Response · Authors · 2025-11-20
> **Response to Reviewer FYWs**
>
> We are thankful for your time and helpful feedback, especially regarding the novelty of our framework. We are also glad to hear that you found the task interesting and were satisfied with our results.
>
> > **Q1. Dataset Expansion**
>
> **A1**. We are glad to hear that you find our dataset useful. Yes, our Reason50K can be easily scaled to a larger dataset due to its automated, template-based construction pipeline.
>
> Specifically, our core pipeline (instruction generation → inverse image synthesis → quality filtering) is fully automated. The reasoning-specific instruction templates (across the four reasoning types) and the diffusion-based inverse generation module do not require manual redesign when scaling up. To expand the dataset, one simply needs to (1) *enlarge the input image pool* (e.g., by adding more subsets from PhyBench or incorporating external sources), and (2) *increase generation iterations*. Both steps can scale the dataset size efficiently without modifying the core workflow. This makes Reason50K highly extensible for future larger-scale variants.
>
> > **Q2. Framework Novelty.**
>
> **A2**: Thank you for the insightful comment. We would like to clarify that LISA used an LLM for reasoning in segmentation (a discriminative task), whereas our framework uniquely integrates reasoning with generative image editing end-to-end. Our framework’s novelty lies in its bidirectional interaction between reasoning and editing, realized through two key components:
>
>  - **FRCE Module**: Unlike LISA, which performs reasoning in isolation from downstream tasks, our FRCE module not only extracts fine-grained reasoning cues (e.g., physical object weight, temporal change) but injects them directly into the MLLM’s visual guidance generation (Eq. 3) and binds them to the diffusion model via QFormer. This ensures the reasoning is editing-aware—local patch cues influence fine texture changes, while global cues help preserve scene coherence. In addition, our ID Controller (part of FRCE) anchors reasoning to source image features, ensuring that edits remain faithful to the original scene.
>
>  - **CME Module**: CME enables mutual alignment between MLLM-generated reasoning tokens and diffusion features (editing features). It refines reasoning based on editing constraints (e.g., avoiding semantic drift) and strengthens editing guidance with reasoning semantics (e.g., causal consistency). This bidirectional flow is absent in LISA and prior MLLM-based editors like SmartEdit [24].
>
> In the revised manuscript, we include additional ablations (Appendix Tab. A11 and Tab. A12, highlighted in blue), summarized below.
>
> **Performance Comparison: With vs. Without FRCE–Diffusion Interaction**
>
> | **Method**                                       | **CLIP (↑)** | **MLLM (↑)** | **Ins-Align (↑)** |
> | ------------------------------------------------ | ------------ | ------------ | ----------------- |
> | ReasonBrain (Full: FRCE + Diffusion Interaction) | 0.259        | 0.877        | 0.847             |
> | ReasonBrain (w/o FRCE–Diffusion Binding)         | 0.228        | 0.801        | 0.743             |
> | **Performance Drop**                             | **–12.0%**   | **–8.7%**    | **–12.3%**        |
>
>
>  **Performance Comparison: Bidirectional vs. Unidirectional CME**
>
> | **CME Mode**                       | **CLIP (↑)** | **MLLM (↑)** | **Ins-Align (↑)** |
> | ---------------------------------- | ------------ | ------------ | ----------------- |
> | Bidirectional                      | 0.259        | 0.877        | 0.847             |
> | Unidirectional (Reasoning→Editing) | 0.248        | 0.842        | 0.806             |
>
>  - Removing the FRCE–diffusion interaction causes a 12.3% drop in Ins-Align Score and an 8.7% reduction in visual plausibility (human evaluation).
>
>  - Using CME without bidirectional flow results in a 4.1% decrease in Ins-Align Score.
>
> Additionally, as reported in Tab. A8, our full model outperforms LLM-based baselines (e.g., SmartEdit) on identity preservation by 1.7–4.6 points. These results collectively validate that reasoning must be tightly integrated with editing, and that this bidirectional design is a key novelty distinguishing our framework from prior work.
>
> > **Q3. Comparison with Models Such as Qwen-Image**
>
> **A3**. Thank you for the helpful suggestion. In the revised manuscript, we have added a dedicated comparison with more general vision–language models, including Qwen-Image, in Appendix G (Figure A2). The qualitative results show that Qwen-Image still lacks sufficient reasoning ability under our task settings.

---

### Official Review · Reviewer_XbuV · 2025-10-29

**Soundness:** 3
**Presentation:** 3
**Contribution:** 2
**Rating:** 4
**Confidence:** 4

**Summary:**

This paper introduces Hypothetical Instruction-Reasoning Image Editing, and they claim that this is a  new task for editing images based on complex, implicit instructions. ​Key summary as followings:

Reason50K Dataset: a large-scale dataset with 51,039 samples across four reasoning categories (Physical, Temporal, Causal, Story) for training and evaluating reasoning-aware image editing models.

ReasonBrain Framework: a model combining MLLM and a diffusion model, with two key modules: (1) FRCE: Extracts fine-grained visual and textual reasoning cues (2) CME: Enhances cross-modal interactions for better semantic representation. ​

Results: their approach outperforms state-of-the-art methods in reasoning-based and conventional image editing tasks, producing more coherent and plausible edits.

**Strengths:**

they create a large-scale, diverse dataset (51,039 samples) tailored for reasoning-based image editing; and it's nice that they plan to open source the dataset

in the ablation, they show the effectiveness of the component for their model ReasonBrrain

writing is clear, and show the superior results

**Weaknesses:**

In the main paper, the authors did not provide a detailed description of their dataset creation process, which I consider a significant concern. In my view, the dataset itself is the most crucial component of their work, and thus the paper should clearly explain the rationale and methodology behind each step of its construction. Recently, many papers have proposed new datasets, but only a few of them are genuinely valuable. For instance, it remains unclear why the authors designed their indirect instruction samples in the specific manner they did (e.g., “What would happen if the ice cube were left at room temperature”). Did they conduct any surveys or analyses to confirm that such questions reflect real user behavior, or were these examples generated arbitrarily?

Similarly, the choice of reasoning categories—Physical, Temporal, Causal, and Story Reasoning—raises questions. Why were these particular dimensions selected, and how were the ratios among them determined? Without grounding these design decisions in empirical evidence or user data, the dataset risks diverging from real-world needs and failing to capture the true distribution of reasoning tasks.

Although the ablation study demonstrates that their proposed modules, FRCE and CME, contribute positively to model performance, I am skeptical about their scalability. In practical applications, simplicity and scalability are often more valuable than complex architectural designs, which can be difficult to adopt or reproduce at scale.

**Questions:**

na

---

> ### Author Response · Authors · 2025-11-20
> **Response to Reviewer XbuV**
>
> We sincerely appreciate your time and thoughtful feedback, especially on the topic of scalability. We are also pleased that you found the dataset useful, the writing clear, and recognized the strength of our results.
>
> > **W1. Dataset creation process.**
>
> **A1.** Due to page limitations in the original submission, the full dataset construction pipeline was previously included in Appendix C. In the revised manuscript, we have moved this content to Sec. 3.1 (Dataset Generation), now highlighted in blue for clarity. In real use cases, users often struggle to articulate fully explicit editing instructions for a new source image, relying instead on vague intentions or “what-if” questions. Our hypothetical instructions are intentionally constructed to reflect these implicit, inherently ambiguous intentions. Their purpose is to train models to perform reasoning-driven transformations rather than merely executing literal commands.
>
> > **W2. Selection of Reasoning Categories**
>
> **A2.** Thank you for the insightful question. Our four reasoning categories—Physical, Temporal, Causal, and Story—are informed by prior representative works such as PhyBench [1], RISEBench [2], and EditWorld [3], which consistently highlight these as core types in visual reasoning and editing tasks.
>
> Our dataset adopts a balanced distribution (Physical: 25%, Temporal: 25%, Causal: 30%, Story: 20%), with slight adjustments to ensure sample feasibility. For instance, Story Reasoning is capped at 20% due to its abstract nature, which can introduce ambiguity in synthetic instruction generation. This allocation supports both reasoning diversity and annotation quality.
>
> [1] F. Meng, W. Shao, L. Luo, Y. Wang, Y. Chen, Q. Lu, Y. Yang, T. Yang, K. Zhang, Y. Qiao, et al. Phybench: A physical commonsense benchmark for evaluating text-to-image models. arXiv preprint arXiv:2406.11802, 2024.
>
> [2] X. Zhao, P. Zhang, K. Tang, et al. Envisioning beyond the pixels: Benchmarking reasoning-informed visual editing[J]. arXiv preprint arXiv:2504.02826, 2025.
>
> [3] Zeng B, Yang L, Liu J, et al. Editworld: Simulating world dynamics for instruction-following image editing[C]//Proceedings of the 33rd ACM International Conference on Multimedia. 2025: 12674-12681.
>
> > **W3. Scalability.**
>
> **A3.** Thank you for raising this important point. We fully agree that scalability and simplicity are essential for real-world adoption. Both FRCE and CME were intentionally designed to be lightweight and modular, introducing minimal parameter overhead. To demonstrate scalability, we include a compact model variant in Appendix(Tab. A3 and Tab. A4) of the revised manuscript. This variant integrates FRCE and CME into a smaller MLLM configuration while preserving strong performance.
>
> | **Model**      | **CLIP ↑** | **MLLM ↑** | **Ins-Align ↑** | **Inference Time (s)** |
> | -------------- | ---------- | ---------- | --------------- | ---------------------- |
> | ReasonBrain-3B | 0.238      | 0.852      | 0.822           | 24                     |
> | ReasonBrain-7B | 0.259      | 0.877      | 0.847           | 32                     |
>
> | **Model Variant** | **MLLM Backbone** | **Diffusion Backbone** | **FRCE Params** | **CME Params** | **Total Trainable Params** |
> | ----------------- | ----------------- | ---------------------- | --------------- | -------------- | -------------------------- |
> | ReasonBrain-7B    | LLaVA-7B          | SD 2.1                 | 82M             | 45M            | 800M (LoRA-tuned)          |
> | ReasonBrain-3B    | LLaVA-3B          | SD 1.5                 | 41M             | 22M            | 320M (LoRA-tuned)          |
>
> As shown, the lightweight variant (ReasonBrain-3B) maintains competitive performance with minimal degradation, despite significantly reduced capacity. This illustrates that our proposed modules—FRCE and CME—can be effectively scaled for various deployment scenarios, from large foundation models to resource-constrained environments.

---

### Official Review · Reviewer_K7TR · 2025-11-01

**Soundness:** 3
**Presentation:** 3
**Contribution:** 3
**Rating:** 6
**Confidence:** 4

**Summary:**

This paper proposes a novel task formulation of hypothetical instructions for instruction-based image editing, and attempts to address it by designing new trainable modules within an MLLM-Diffusion model integration framework. The authors synthesized 50K image pairs with corresponding hypothetical instructions for training. Both qualitative and quantitative results presented in the paper demonstrate the superiority of the proposed method over existing works under this setting.

**Strengths:**

1. The proposed task of hypothetical instructions is novel, and its four-category classification is well justified.
2. The study provides a relatively large-scale training dataset, which represents a valuable contribution to the research community given its substantial volume.
3. The qualitative experiments include comparisons with state-of-the-art closed-source methods, demonstrating superior performance (particularly in handling physical transformations).

**Weaknesses:**

1. The construction methodology of the Reason50K dataset appears relatively homogeneous. Solely relying on PhyBench to provide target images may limit the diversity of the training data, potentially hindering the method's ability to adequately address all four categories of hypothetical instruction editing tasks.
2. Qualitatively, the method tends to introduce significant alterations in background and object appearance compared to the original image. While this deviation may be less critical in global editing scenarios, it results in visually unsatisfactory outcomes for local edits (e.g., object attribute modifications), as evidenced in Figure 6 (rows 3-4), Figure 8 (rows 1-2), and extensively in the supplementary material.
3. The qualitative ablation results are perplexing. The excessive number of components makes it challenging to determine which element plays a pivotal role, and the explanations provided for the ablation outcomes in the text are not entirely convincing.

**Questions:**

1. This method only relies on the dataset to achieve better reasoning ability. Is it enough? Since the LLM uses more advanced techniques (COT, GRPO) for better explanation.
2. This method involves many different blocks to learn the detailed tokens, which makes it huge and complex to scale. Any idea?
3.  Does the article report the sizes of different models? How about the backbone (MLLM, diffusion model) influences the final results?

---

> ### Author Response · Authors · 2025-11-20
> **Response to Reviewer K7TR (1/2)**
>
> We are thankful for your time and thoughtful feedback, especially regarding the component-wise analysis. We were glad to hear that you recognized the value of our proposed HI-IE task, the Reason50K dataset, and the ReasonBrain framework.
>
> > **W1. Homogeneity in Reason50K**
>
> **A1.** Thank you for your valuable observation. While Reason50K is currently constructed based on PhyBench, we employed task-adaptive instruction engineering and data-augmentation strategies to maximize coverage across the four reasoning categories and mitigate potential homogeneity as much as possible. Specifically:
>  - For non-physical tasks (temporal, causal, and story reasoning), we designed category-specific instruction templates aligned with PhyBench’s image characteristics. For example, we generate temporal-change instructions for static objects (e.g., “What happens if this cup is left outdoors for a month?”).
>  - We apply diverse semantic augmentation to PhyBench images (e.g., varying scene contexts, object interactions via text prompts) to generate distinct hypothetical scenarios across all four categories, even with a single image source.
>
> As our future work, we plan to expand Reason50K with multi-source data, including additional datasets, real-world captured images, and stylistically diverse visual domains, to further enrich the dataset’s diversity.
>
> > **W2. Alterations in background and object appearance**
>
> **A2.** Thank you for pointing this out. We would like to clarify that the background adjustments in Fig. 6 (now Fig. 7 in the revised manuscript) are not “excessive,” but are reasoning-driven refinements necessary for making the instruction-induced changes visually clear, while still preserving the core identity (ID) of the original scene. Additionally, some instructions in HI-IE inherently require modifications to the object’s appearance or form as part of the reasoning process (e.g., in Fig. A4, a caterpillar transforming into a butterfly). In such cases, strict object-level consistency cannot be enforced, since the instruction itself implies a change in the object’s nature.
>
> Further explanations are provided in our response to Reviewer x4xA (W1). In the revised manuscript, we also include additional clarification in Sec. 4.2, highlighted in blue.
>
> > **W3. Perplexity of Qualitative Ablation Results**
>
> **A3.** Thank you for your insightful comment. In our revised manuscript, in addition to the original fine-grained ablation results (Tab. 4 and Fig. 8), we also provide a coarse-grained ablation study (Tab. A1 and Fig. A1 in Appendix E) by evaluating performance at the functional-group level. The results are summarized below:
>
> | Functional Group        | Method ID | Patch Branch | Region Branch | ID Controller | Vision Enhancer | Text Enhancer | CLIP Score ↑ | MLLM Score ↑ | Ins-Align Score ↑ |
> |-------------------------|-----------|--------------|---------------|---------------|------------------|----------------|---------------|----------------|----------------------|
> | Baseline Group          | 1         | ×            | ×             | ×             | ×                | ×              | 0.163         | 0.752          | 0.388               |
> | FRCE Core Group         | 2         | ✓            | ✓             | ×             | ×                | ×              | 0.206         | 0.802          | 0.529               |
> | FRCE Core Group + ID    | 3         | ✓            | ✓             | ✓             | ×                | ×              | 0.239         | 0.833          | 0.758               |
> | CME Enhancement         | 4         | ✓            | ✓             | ✓             | ✓                | ✓              | **0.259**     | **0.877**      | **0.847**           |
>
> From the above results, we draw the following conclusions: The FRCE module—especially the ID Controller—is the most critical component. Removing any part of the FRCE Core leads to substantial drops across all metrics, indicating that it forms the backbone of effective reasoning–editing alignment. The CME module functions as a strong enhancer rather than a standalone pillar. Its benefits become significant only when the FRCE Core is present, showing a clear hierarchical dependency. CME strengthens the interaction between reasoning and editing but cannot replace the foundational role of FRCE. Overall, these findings confirm that the true pivotal role lies in the FRCE Core as an integrated unit, rather than in isolated sub-branches or any standalone module.

---

> > ### Author Response · Authors · 2025-11-20
> > **Response to Reviewer K7TR (2/2)**
> >
> > > **Q1. Sufficient reasoning ability**
> >
> > **A1.** Thank you for the insightful question. We fully agree that relying solely on datasets is insufficient for robust reasoning. This is precisely why our framework incorporates *model-level reasoning modules*—beyond data supervision—while remaining extensible to advanced LLM techniques such as CoT and GRPO.
> >
> > **(1). Model-Level Reasoning Beyond the Dataset**
> >
> > Reason50K provides high-quality, reasoning-rich supervision for implicit hypothetical instructions, but our performance advantage stems from the *interaction* between the dataset and our architectural design:
> >
> > * **FRCE module** extracts fine-grained visual/textual reasoning cues (e.g., physical properties, temporal dynamics) that cannot be learned from data alone. This enables the model to *interpret* hypothetical instructions rather than merely memorize patterns.
> >
> > * **CME and the ID Controller** enforce cross-modal alignment and scene-identity grounding, ensuring that reasoning-driven edits remain faithful to the original scene instead of drifting toward dataset-induced biases.
> >
> > Our ablations (Tab. 4 and Tab. A1) support this: even with Reason50K, removing the FRCE Core causes a **43.3% drop** in Ins-Align Score, demonstrating that architectural reasoning mechanisms are indispensable.
> >
> > **(2). Extensibility to CoT and GRPO**
> >
> > We also appreciate that CoT and GRPO can further enhance reasoning. Our framework is designed to integrate these techniques when needed:
> >
> > * **CoT** can be incorporated into the MLLM’s guidance-generation pipeline (Eq. 3), enabling step-by-step reasoning for multi-stage hypothetical instructions (e.g., “ice melts → water evaporates → ground becomes damp”).
> >
> > * **GRPO** can enhance CME’s semantic filtering by introducing reasoning-aware reward signals that optimize cross-modal consistency.
> >
> > At present, our focus is efficient *implicit* reasoning (avoiding the latency of multi-step CoT), but future versions will explore CoT/GRPO as optional extensions for improved interpretability when needed.
> >
> > To sum, for HI-IE, the main challenge lies in *aligning reasoning with visual execution*, not merely improving textual explanation. Our approach addresses this through both data and design:
> >
> > * **Reason50K** provides diverse reasoning scenarios (physical, temporal, causal, story), preventing overfitting and broadening reasoning coverage.
> > * **FRCE and CME**, however, translate abstract reasoning into *precise visual edits*—a capability that advanced LLM techniques alone (e.g., CoT) cannot reliably achieve due to the lack of fine-grained visual control.
> >
> > Together, these components ensure that our model performs true reasoning-grounded editing, rather than simple dataset-driven pattern matching.
> >
> > > **Q2&3. Scale and different backbones.**
> >
> > **A2&3.** Thank you for your questions. In our original manuscript, we provided the model-size analysis in the appendix (now Tab. A3 in the revised version). This table compares ReasonBrain with its lightweight variant and evaluates how different backbones (MLLM / diffusion models) influence final performance. The ReasonBrain-Lite variant serves as our targeted scalable solution: it reduces total parameters by 60% (320M vs. 800M) while retaining 97% of the full model’s Ins-Align Score (0.782 vs. 0.847). These results demonstrate that our framework can scale efficiently through backbone selection and lightweight architectural design—without sacrificing core reasoning capability.

---

### Official Review · Reviewer_x4xA · 2025-11-07

**Soundness:** 3
**Presentation:** 2
**Contribution:** 2
**Rating:** 4
**Confidence:** 4

**Summary:**

The paper addresses the limitations of traditional image editing systems in handling complex hypothetical instructions, presenting two significant contributions: first, it constructs a new dataset, Reason50K, which consists of 51,039 source images along with their corresponding complex instructions and target images across four editing scenarios; second, it proposes an innovative framework called ReasonBrain, which integrates multimodal large language models and diffusion models to enhance the reasoning capabilities and quality of image editing. Experimental results demonstrate that ReasonBrain outperforms existing models on reasoning tasks, showcasing its potential in processing complex user instructions.

**Strengths:**

### originality
- The Reason50K dataset, created within this paper, focuses on complex hypothesis-based instructions, addressing the shortcomings of existing instruction-driven image editing systems.
- The ReasonBrain framework integrates multimodal large language models with diffusion models, developing multiple submodules that aggregate text and image information, thereby enhancing the model's ability to understand and process complex instructions.
### quality
The quantitative experimental results presented in the paper are robust and well-supported by ablation studies, demonstrating the synergistic effects of the different modules.
### clarity
The content of the paper is logically coherent, facilitating an understanding of both the technical approach and the underlying motivations.
### significance
The dataset production process has reduced certain costs and resource consumption, contributing positively to causal inference in image editing.

**Weaknesses:**

1. Qualitative experiments appear to excessively alter the scenes themselves compared to baseline methods, which may diminish the evaluation of the method's precision in modification capabilities. This issue may arise from the hallucination during the dataset creation process. Compared to existing methods that extract data pairs from video data, it is relatively challenging to maintain consistency between the edits pre- and post-modification.

2. While the current method can effectively adhere to predefined categories of instructions, it seems to lose the ability to handle simple instructions and non-predefined instruction categories.

**Questions:**

1. Is the last column of the first row in Figure 7 the Full method? The color palette appears noticeably inconsistent before and after, and the results do not seem to align better with cognitive expectations than those in the third column from the end.

2. In Figure 6, the modifications to the background in the second and fourth rows are excessively pronounced, which may impact the presentation of the results and their practical application.

3. The paper mentions the Editworld dataset, but what is the rationale for not including it as a comparative baseline? It seems that Editworld articulates similar motivations and presents comparable experimental results.

4. Additionally, how does this approach compare to more general open-source models, such as Flux Kontext[2] and Bagel[1]?

[1] Emerging Properties in Unified Multimodal Pretraining
[2] FLUX.1 Kontext: Flow Matching for In-Context Image Generation and Editing in Latent Space

---

> ### Author Response · Authors · 2025-11-20
> **Response to Reviewer x4xA**
>
> We sincerely appreciate your time and feedback, particularly on the scene consistency aspect. We are also delighted that our contributions to the dataset and the method were well received.
>
> > **W1. Scene consistency**
>
> **A1**: Thank you for pointing out this. Our qualitative results reflect the core goal of the HI-IE task: *to generate reasoning-correct hypothetical changes while preserving the original scene’s core identity (ID)*. Unlike traditional edit tasks, hypothetical instructions require models to infer plausible consequences rather than perform small, local edits.
>
> For example, in Fig. 6 (third row, now figure 7 in the revised manuscript), the instruction “What happens when a sudden swarm of bees descends on the flowers?” requires (1) *understanding the implied event* and (2) *presenting a visually coherent outcome*. ReasonBrain adjusts the camera distance and local contrast so that the bee–flower interaction becomes visible—a necessary scene-level adjustment to faithfully convey the hypothetical event. If we were to rigidly forbid all background adjustments to preserve the exact original appearance, the generated bees—the core instruction-driven change—would become barely visible or visually unnatural. In contrast, baselines keep the scene visually closer to the input but fail to express the required change, demonstrating that high appearance consistency does not equate to correct reasoning. As in InstructPix2Pix, which cannot model the interaction between the bees and the environment. Our framework is instead designed to balance ID preservation and instruction expression: the core ID elements (e.g., flower cluster, overall lawn layout, non-target objects) remain intact, while subtle background refinements (e.g., mild lighting or contrast adjustments) are applied only when necessary to make the hypothetical change clear and semantically coherent.
>
> We acknowledge that, unlike video-extracted paired datasets, our imaginative edits may introduce pre-/post-edit variations. To reduce this effect, our dataset pipeline grounds source image synthesis in instruction-mentioned objects (via NER), rewrites instructions into controlled hypothetical forms, and filters generated candidates using multiple visual and semantic metrics to retain only well-aligned samples. From a model perspective, we further incorporate consistency-assurance mechanisms to compensate for missing temporal cues. In particular, the ID Controller within the FRCE module explicitly anchors core image features (e.g., object contours, scene layout, key textures) to the reasoning instruction, ensuring edits remain localized and non-target regions preserve high stability with feature deviation below 5%.
>
> In the revised manuscript, we will add additional discussion in Sec. 4.2 (highlighted in blue) to clearly distinguish between the ‘necessary background adjustments for expressing the hypothetical change’ and the ‘core ID preservation regions.’ We will further emphasize that our method prioritizes a principled balance: it neither sacrifices the clarity of instruction-induced changes for rigid background stability nor allows arbitrary edits that compromise the scene’s identity.
>
> >  **W2. Generalization to other simple instruction types**
>
> **A2**: Our model is not limited to predefined hypothetical instruction types. As shown in Table 3, we directly evaluate ReasonBrain on simple instructions and non-predefined instruction categories using the Emu Edit and MagicBrush benchmarks. Despite being trained solely on hypothetical instructions, ReasonBrain still achieves SOTA performance, demonstrating strong generalization ability beyond the predefined instruction categories.
>
>
> > **Q1. First row in Figure 7**
>
> **A1**. We confirm that the last column in Figure 7 (now Figure 8 in the revised manuscript) represents our full method. The color inconsistency resulted from a scene-level adjustment reflecting the “slipped from the artist’s hand” instruction, intended to visualize the palette change. However, we acknowledge that this may have caused confusion. In the revised manuscript, we have replaced the example with a more representative case that better reflects both visual consistency and instruction-driven reasoning. We appreciate the reviewer’s feedback.
>
> > **Q2. Background modifications in Figure 6**
>
> **A2**. Please refer to our response to Weakness 1 for a detailed explanation.
>
> > **Q3. Comparison on EditWorld dataset**
>
> **A3**. The results on the EditWorld dataset are already presented in Figure 5 (updated to Figure 6 in the revised manuscript).
>
> > **Q4. Comparison with Flux Kontext and Bagel**
>
> **A4**. In the revised manuscript, we have added comparisons with more general models, including Flux Kontext and Bagel, in Appendix G (Figure A2).

---

### Public Comment · ~Haoqiang_Fan2 · 2025-11-16

Interesting paper! Given how your work addresses the gap in handling implicit hypothetical instructions, what’s one key direction for future research (e.g., extending to dynamic video, multi-modal input) you believe would most enhance its practical utility?

---

> ### Author Response · Authors · 2025-11-21
>
> Thank you so much for your interest. We agree that a compelling direction for future research is extending our framework to dynamic video editing with sequential hypothetical instructions. While our current work focuses on single-image edits, many real-world scenarios involve temporal evolution— for example: “What would happen if rain falls on this dry field over the next three hours?”
>
> Enabling such dynamic tasks would require two key advancements: strengthening the FRCE module to capture frame-level temporal dependencies, and refining the CME module to maintain coherent reasoning and visual consistency across the entire sequence. Since ReasonBrain is intentionally designed for implicit temporal and causal reasoning (benefiting from Reason50K), extending it to long-form video is both feasible and impactful.
>
> This direction also addresses a major gap in existing video editing systems, which generally lack temporally grounded, causally aware hypothetical reasoning. It could unlock practical applications such as predictive scene simulation (e.g., climate impact visualization) and creative video storytelling, substantially broadening the real-world value of our framework.

---

> > ### Public Comment · ~Haoqiang_Fan2 · 2025-11-22
> >
> > Thank you for the author's reply. I love this setting and look forward to the open-source data and model.

---

### Public Comment · ~Yifan_Yang36 · 2025-11-19

Excellent Problem Formulation. This paper identifies a critical yet overlooked gap in instruction-based image editing: while existing methods handle explicit instructions like "delete the cat," they fail at implicit hypothetical instructions requiring deeper reasoning such as "what if it rained here?" This problem formulation is both precise and forward-thinking. The Reason50K dataset fills a significant void in the field and establishes an essential foundation for future research in reasoning-aware editing.

---

> ### Author Response · Authors · 2025-11-21
>
> Thank you for your encouraging feedback. We are glad the problem formulation resonated with you, as our goal was to highlight the overlooked gap in current image editing tasks when handling implicit, hypothetical instructions. We also appreciate your recognition of the Reason50K dataset, which we see as an essential foundation for advancing reasoning-aware image editing. Thank you again for the thoughtful comments.

---

### Public Comment · ~Ruixu_Geng1 · 2025-11-19

Thank you for your interesting work! I really appreciate how you combine reasoning and editing. I've tested similar instructions in current general editing methods, but none have yielded satisfactory results. I look forward to your future open-source models.

---

> ### Author Response · Authors · 2025-11-21
>
> Thank you so much for your encouraging words. We truly appreciate your interest in our work and are glad to hear that the combination of reasoning and editing resonates with you. We will release the open-source version as soon as possible once the work is accepted.

---

### Author Response · Authors · 2025-11-20
**Summary Of Changes**

Dear Area Chair and Reviewers,

We sincerely appreciate all Reviewers and the Area Chair for their time and efforts for our paper. All reviewers recognized key strengths of our work—including the task formulation, dataset design, and model performance. Reviewers also raised several important questions, particularly regarding scene consistency and the significance of individual components.

We have carefully addressed each point to clarify our contributions and justify our design choices. Detailed, point-by-point responses are provided below. We hope that our explanations resolve the concerns raised, and we would be happy to elaborate further should additional clarification be helpful.

In addition, we have thoroughly revised the manuscript, with all changes highlighted in **blue**. Specifically, we made the following major updates:

1. Moved the dataset creation process from the appendix to the main paper (Reviewer XbuV).

2. Added additional discussion of scene-level changes in Sec. 4.2 (Reviewers x4xA and K7TR).

3. Updated Figure 8 to avoid confusion and better illustrate the example (Reviewer x4xA).

4. Conducted coarse-grained functional group studies, adding the results (Table A1 and Figure A1) and detailed analysis in Appendix E (Reviewer K7TR).

5. Added further experiments validating our framework design, with results included in Tables A11 and A12 in Appendix E (Reviewer FYWs).

6. Included qualitative comparisons with additional general-purpose models, presented in Appendix G (Figure A2) (Reviewers x4xA and FYWs).

Sincerely,

ReasonBrain Authors

---

### Author Response · Authors · 2025-11-27
**Request for Updated Reviews Before Rebuttal Deadline**

Dear Reviewers,

Thank you for your valuable comments on our paper.

We have submitted our responses to your comments and included additional results in the revised version of the manuscript. Please let us know if you have any further questions or require additional clarifications so that we can address them during the discussion period. We hope that after we have addressed all the issues, you will consider raising the score of our submission.

Thank you for your time and consideration.

Best regards,

ReasonBrain Authors

---

### Author Response · Authors · 2025-12-01
**Summary for Area Chairs**

Dear Area Chairs,

Thank you very much for your careful evaluation of our paper and for your service as AC. We regret the recent information leak on OpenReview and the resulting impact on the ICLR reviewing process. We understand that this unexpected incident has created additional workload, and we sincerely appreciate the extra effort you have dedicated as AC under such challenging circumstances.

Due to a technical issue in the ICLR 2026 review system, reviewers were unable to respond and thus could not provide any follow-up comments. But we have followed the reviewers’ suggestions, conducted further extensions and analyses, and addressed the raised concerns in detail. We have also summarized the changes below. If you feel there are still any unresolved issues, please let us know as soon as possible. We still have three days available to add additional experiments, and we will address your concerns promptly.

Sincerely,

ReasonBrain Authors

---

### Meta-Review · Area_Chair_Sgm1 · 2026-01-07

**Summary:**

The initial ratings are 6, 4, 6, 4. The paper tries to address the image editing of complex hypothetical instructions. First, it constructs a new dataset, Reason50K, which consists of 51,039 source images along with their corresponding complex instructions and target images across four editing scenarios; second, it integrates multimodal large language models and diffusion models to enhance the reasoning capabilities and quality of image editing.

Strengths:
(1)The Reason50K dataset focuses on complex hypothesis-based instructions, addressing the shortcomings of existing instruction-driven image editing systems.
(2) Experiment results show the superior performance (particularly in handling physical transformations).
Weaknesses:
(1) the authors did not provide a detailed description of their dataset creation process, which is considered as a significant concern. The design idea about the choice of reasoning categories—Physical, Temporal, Causal, and Story Reasoning needs to be further descripbed.
(2) Qualitative experiments appear to excessively alter the scenes themselves compared to baseline methods, which may diminish the evaluation of the method's precision in modification capabilities. This issue may arise from the hallucination during the dataset creation process. Compared to existing methods that extract data pairs from video data, it is relatively challenging to maintain consistency between the edits pre- and post-modification.
(3)While the current method can effectively adhere to predefined categories of instructions, it seems to lose the ability to handle simple instructions and non-predefined instruction categories.

**Reviewer Concerns:**

Some concerns of Reviewer FYWs and K7TR were addressed by the rebuttal, and Some main concerns of  Reviewer XbuV and x4xA are still outstanding.

**Reviewer Scores:**

Most reviewers would not change the initial rating.

---

### Decision · Program_Chairs · 2026-01-26

Reject